# FROM ATOM TO SPACE: A REGION-BASED READOUT FUNCTION FOR SPATIAL PROPERTIES OF MATERIALS

**Jiawen Zou[1], Zhongyao Wang[1,2], Hao Qi[1], Wemin Tan[1,3,†], Bo Yan[1,†]**
[1]College of Computer Science and Artificial Intelligence,
Shanghai Key Laboratory of Intelligent Information Processing,
Fudan University, Shanghai, China
[2]Shanghai Innovation Institution, Shanghai, China
[3]Shanghai Academy of AI for Science, Shanghai, China
[†]Corresponding author

## ABSTRACT

The message passing–readout framework has become the de facto standard of graph neural networks (GNNs) for material property prediction. However, most existing readout functions are built on an atom-decomposable inductive bias, i.e. the material-level property or feature can be reasonably assigned to contributions of individual atoms. This is a strong bias and may not hold for all properties, limiting the application scenarios (e.g. gas adsorption or separation of Metal Organic Frameworks, MOFs). In this work, we propose a region-based decomposition perspective, reformulating material properties as integrals over space and pooling contributions from spatial regions rather than atoms. Specifically, we propose a novel readout function named SpatialRead. SpatialRead introduces additional spatial nodes to represent a voxelized space, transforming the atomic isomorphic graph into a heterogeneous atom–space graph with unidirectional message flow from atoms to spatial nodes. To combine the two types of inductive bias, multi-modal methods can be used to fuse the features of atoms the spatial nodes. Such a region-based readout function is especially suited for spatial properties such as gas adsorption capacity, separation ratio. Extensive experiments demonstrate that a simple PaiNN–Transformer-based SpatialRead trained from scratch outperforms state-of-the-art pre-trained foundation models on these special tasks. Our results highlight the importance of designing physically grounded readout functions tailored to the target property. The code and dataset can be found in github `https://github.com/nankusa/SpatialRead`.

## 1 INTRODUCTION

The field of material artificial intelligence is fundamentally shaped by the task of material property prediction. Accurate predictions can significantly accelerate the screening and design of novel materials by bypassing costly and time-consuming experiments. In this domain, Message Passing Neural Networks (MPNNs) have emerged as the state-of-the-art paradigm for both property prediction and material generation. MPNNs represent a material as a graph, with atoms as nodes and edges connecting neighboring atoms. A typical MPNN consists of two stages: (1) message passing, where node features are iteratively updated through local aggregation, and (2) readout, where the final node features are aggregated into a graph-level property.

Readout is a critical component of this architecture. Simple pooling functions, such as global sum or mean, have proven remarkably successful and even scaled to foundation models with hundreds of millions of training samples (Gasteiger et al., 2020; 2021; Shoghi et al., 2024). More complex readout functions, such as GraphTrans (Wu et al., 2021) and GMT (Baek et al., 2021), introduce architectural sophistication but ultimately still treat nodes as the fundamental units of aggregation. These designs reflect the implicit node (atom)-decomposable inductive bias: a graph-level property or feature can be decomposed into node contributions. While it works well for many tasks, its broader applicability has not been carefully examined.

A typical counterexample of the atom-decomposable inductive bias arises in porous materials such as metal–organic frameworks (MOFs). Being promising to applications such as gas adsorption and clean energy storage (Snyder et al., 2023; Nugent et al., 2013; Datta et al., 2015; Zhao et al., 2018; Yang et al., 2012; Zhou et al., 2022), these materials attract broad research interest. One of the key properties of such material - the gas adsorption capacity - can naturally be expressed as the summation of the adsorption capacities of all regions within the material. Other similar examples include accessible pore volume, adsorption heat, gas selectivity ratio, etc. In these cases, the target property, which we named as *Spatial Properties* should be decomposed into contributions of each spatial region instead of atom. The inherent mismatch between these properties and node-decomposable readouts highlights a critical blind spot in current graph learning methods for materials.

To correctly consider such region-based inductive bias, we propose a novel perspective to reformulate the node-decomposable property as the integral over space. We first prove the equivalence of the two expressions. Thus the reformulation is simply a kind of inductive bias without harming the expressivity of the model. Based on this, we propose a novel readout module named SpatialRead. The central idea is introducing spatial nodes that represent voxelized space. This transforms the conventional atomic graph into a heterogeneous atom–space graph. Messages are passed unidirectionally from atoms to spatial nodes. However, pooling features from spatial nodes may compromise the performance on those non-spatial properties. To make the network adaptively select the correct inductive bias of atomic and space-decomposition, We use attention-based multimodal method to process both atom and spatial node features. This approach enables SpatialRead to maintain comparable performance in non-spatial attributes, thus providing a general flexible option.

SpatialRead is especially suited for spatial properties such as gas adsorption capacity without harming the performance on other tasks such as the MatBench dataset (Dunn et al., 2020). To evaluate the performance for spatial properties, we collect and release a dataset, covering 4 porous materials and 27 downstream tasks. Remarkably, on these special properties, a simple PaiNN–Transformer architecture equipped with SpatialRead, trained from scratch, outperforms a GemNet-based foundation model pretrained on 120 million samples (Shoghi et al., 2024). In summary, our contributions are:

- We reformulate the node-decomposable property as the integral over space and prove the equivalence. Such perspective provides a new inductive bias that the target property can be decomposed into contributions of each spatial region instead of atom.

- We propose SpatialRead, which uses spatial nodes to construct heterogeneous atom–space graph and then use multimodal method to achieve adaptive selection of the two kinds of inductive bias. SpatialRead achieves the state-of-the-art performance on spatial properties without pre-training and maintains comparable performance on non-spatial properties.

- We release a collected benchmark, covering 4 porous materials and 27 downstream tasks.

## 2 Preliminary and Related Works

Message Passing Neural Networks (MPNNs) are first formally defined by Gilmer et al. (2017) and have since achieved great success in fields like chemistry and material science. In these domains, a molecule or a material is represented as a graph $G = (V, E)$, where $V$ is the set of nodes (atoms) and $E$ is the set of edges. Each node $v_i \in V$ is typically represented by its atomic number $x_i$ and coordinates $pos_i$. Edges are usually constructed based on a distance cutoff $r_{cut}$ between adjacent atoms, often with an additional limit on the maximum number of neighbors to ensure computational efficiency.

An MPNN consists of two main stages: a *message passing* process and a *readout* process. In the message passing process, each atom updates its own feature vector by aggregating "messages" from its neighbors. For a typical $T$-layer MPNN, this process can be formally expressed as:

$$h_{v_i}^{t+1} = U_t(h_{v_i}^t, \{h_{v_j}^t, e_{v_i,v_j}^t\}_{v_j \in \mathcal{N}(v_i)}) \tag{1}$$

where $h_{v_i}^t$ is the feature vector of atom $v_i$ at layer $t$, $\mathcal{N}(v_i)$ denotes the set of its neighbors, $e_{v_i,v_j}$ represents the edge feature between $v_i$ and $v_j$, and $U_t$ is the node feature update function of the $t^{th}$

layer neural network. More advanced MPNNs may also incorporate edge updates, as seen in models like DimeNet (Gasteiger et al., 2020) and GemNet (Gasteiger et al., 2021), which use angular and dihedral information to enrich edge features. After $T$ layers of message passing, each node possesses a rich representation vector $h_{v_i}^T$. The final step is the readout process, which pools these node features to generate a graph-level property. The two most common readout strategies are:

$$\text{Feature-level:} \qquad h_{graph} = \text{Pool}(\{h_{v_i}^T\}_{v_i \in V}), \qquad p = \text{MLP}(h_{graph}) \qquad (2)$$

$$\text{Numeric-level:} \qquad o_i = \text{MLP}(h_{v_i}^T), \qquad p = \text{Pool}(\{o_i\}_{v_i \in V}) \qquad (3)$$

Since the numeric value can be regarded as a one-dimension feature, we will not distinguish between two pooling methods hereafter. Although a variety of alternative readout functions exist, such as Set2Set (Vinyals et al., 2015) and diffpool (Ying et al., 2018), most state-of-the-art methods like DimeNet (Gasteiger et al., 2020), GemNet (Gasteiger et al., 2021), ViSNet Wang et al. (2024), and JMP (Shoghi et al., 2024) still rely on these simple pooling forms for most tasks.

The main requirement of the readout function is permutation invariance, i.e., the output should be independent of the input order of atomic features. Formally, $p = readout(\{h_{v_i}^T\}), v_i \in V$. Existing readout methods can be broadly classified into three categories: (1) Flat Pooling Methods directly pool all atom features, such as GMT (Baek et al., 2021) and GraphTrans (Wu et al., 2021) which use attention mechanisms similar to the [CLS] token. (2) Node Clustering Pooling Methods divide nodes into sets and pool hierarchically, including DiffPool (Ying et al., 2018), MinCutPool (Bianchi et al., 2020), and SEP (Wu et al., 2022). Recent advances include Cluster-wise Graph Transformer (Huang et al., 2024) for cluster-node feature interaction, ORC-Pool (Feng & Weber, 2024) based on Ricci Flow, K-MIS-Pool (Bacciu et al., 2023) for topology-preserving downsampling, and GPN (Song et al., 2024) for automatic pooling structure design. (3) Node Drop Pooling Methods select nodes to construct subgraphs, such as TopKPool Gao & Ji (2019); Cangea et al. (2018); Knyazev et al. (2019) which selects top-K nodes by attention scores. Quan et al. (2024) combines TopK selection with clustering for protein representation. SSMA (Keren Taraday et al., 2024) treats neighbor features as 2D discrete signals for enhanced feature mixing, while Xu et al. (2018a) employs jumping-knowledge connections (Xu et al., 2018b) to strengthen graph-level features.

## 3 SPATIAL READ

### 3.1 EMPIRICAL MOTIVATION: IMPLICIT REGION-BASED BEHAVIOR IN STANDARD GNNS

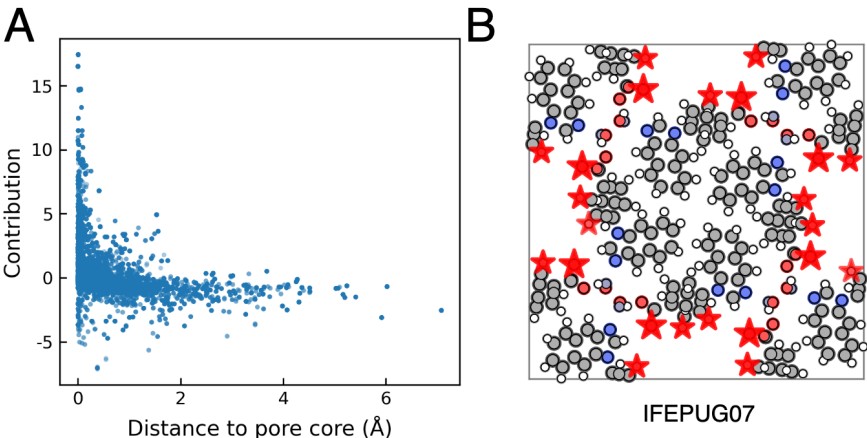

Figure 1: **Contribution of atoms for the adsorption capacity predicted by GNN. (A) Scatter plot between contribution and distance to the nearest pore.** 100 materials from the test set from the CoREMOF dataset. Each dot represents an atom. The horizontal axis shows the distance of the atom from the nearest pore, while the vertical axis indicates the contribution of the atom to the target property. **(B) Visualization of the IFEPUG07 in the CoREMOF dataset.** Top 5% high-contribution atoms are marked in red star.

Before introducing spatial nodes, we examine how conventional atom-based GNNs behave when predicting spatial properties such as gas adsorption in MOFs. To investigate this, we trained a standard PaiNN model on MOF adsorption tasks and computed per-atom contributions using the scalar outputs of its readout (see equation 3). We then compared these contributions with each atom's distance to the nearest pore. Fig. 1 A demonstrate the contribution versus distance to the pore (experiment details can be found in Appendix A.6). We observed a strong alignment between high contributions and pore-adjacent atoms. Among the top 1% of atoms with the highest contribution rate, 86% of them are located within 0.05 angstroms of the pores. Visualization in Fig. 1 B demonstrates a case study, showing that the model attributes most predictive weight to atoms lining the pore channels. These results indicate that when predicting properties that are clearly spatially decomposable, GNNs implicitly learn the key regions and the surrounding atoms. Introducing direct representations of regions may reduce the learning burden of the network and thereby lead to performance improvement.

## 3.2 FROM NODE DECOMPOSITION TO SPATIAL INTEGRATION

Given that atom types and positions fully determine a material structure, the property of a material $G = (V, E)$ can be regarded as a function of its vertices $V$. A typical MPNN applies a local description function $c$ with a limited receptive field: the feature of an atom is determined solely by atoms within a finite radius. To obtain a material-level representation, these atom-level features are aggregated. Formally,

$$h_{\text{graph}} = \sum f(h_{v_i} \mid H), \tag{4}$$

$$h_{v_i} = c(\{v_j\}), \qquad v_j \in \mathcal{N}(v_i), \tag{5}$$

where $f$ is the readout function and $H = \{h_{v_i} \mid v_i \in V\}$ is the set of node features. Here, we reformulate the graph-level feature $h_{\text{graph}}$ as an integral over the continuous spatial domain. To this end, we introduce a contribution function $g(\mathbf{r} \mid \mathcal{S})$, where $\mathcal{S}$ denotes the material structure. Similar to $c$, the function $g$ also has a limited receptive field. The graph-level representation can then be written in the integral form

$$h_{\text{graph}} = \int g(\mathbf{r} \mid \mathcal{S}) \, d^3\mathbf{r} = \int g(\mathcal{N}(\mathbf{r})) \, d^3\mathbf{r}. \tag{6}$$

We refer to properties that admit such a region-based representation as *spatial properties*.

**Definition 3.1.** *A property $p$ is called a* spatial property *if it can be expressed as a functional of a contribution function $g(\mathbf{r} \mid \mathcal{S})$, where $\mathcal{S}$ is the material structure. For instance, gas adsorption capacity is a spatial property: one may define $g(\mathbf{r})$ as the density of adsorbed gas molecules at position $\mathbf{r}$, and the total adsorption capacity is then given by the spatial integral of $g(\mathbf{r})$.*

Despite this formulation, spatial properties are not fundamentally different from other properties in terms of their mathematical or neural representability.

**Theorem 3.1.** *If the readout function $f$ has a limited receptive field, the formulations in equation 4 and equation 6 are equivalent in expressivity. That is, any target property expressible by equation 4 can also be expressed by equation 6, and vice versa.*

A proof is provided in Appendix A.1.1. Theorem 3.1 ensures that under a limited receptive field, reformulating the graph-level representation as a spatial integral does not reduce neural expressivity. When $f$ has an infinite receptive field, the graph-level feature in general cannot be obtained via a simple integral, and a neural architecture with a global receptive field must be applied directly to the contribution function $g(\mathbf{r})$. A detailed discussion is given in Appendix A.1.2.

## 3.3 MODEL ARCHITECTURE

The core idea of our method is to correctly decompose the target property into contributions of discrete spatial regions. To achieve this, we first discretize the spatial property $p$, which is defined as an integral over a continuous domain, into a summation over a set of discrete regions.

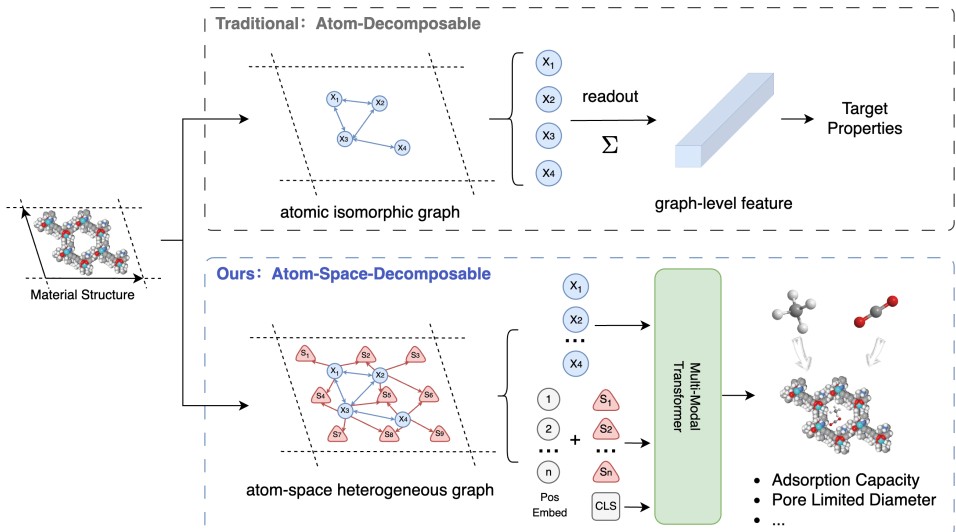

Figure 2: **Architecture of SpatialRead.** The top part represents the traditional method, while the bottom part shows SpatialRead. The material structure is taken as input. For clarity, the lattice vector and boundary is drawn. The traditional method constructs an atom-based isomorphic graph for message passing, and then pools the atomic features to obtain material-level features. In contrast, SpatialRead first uniformly sample spatial nodes (the red triangle nodes) within the lattice. The atom nodes (the blue round nodes) and spatial nodes form a heterogeneous graph. In the message passing process, messages flow between atom nodes (blue lines) and from atom nodes to spatial nodes (red lines). Note there are no messages from spatial nodes to atom nodes. The heterogeneous graph neural network produce an unordered feature list of atom nodes and ordered feature list of spatial nodes. Position embedding can be added to the feature list of spatial nodes. Finally, taking the atom node features as "memory", a decoder of Transformer are used to process both feature lists (Vaswani et al., 2017). An additional [CLS] token is added to the sequence of spatial nodes (Devlin et al., 2019), which can be used to predict spatial properties such as adsorption capacity, separation ratio, or other properties like topology type and pore limited diameter.

$$p = \int_{\mathbb{R}^3} g(\mathbf{r})d^3\mathbf{r} \approx \sum_{j=1}^{N_s} g(\mathbf{r}_j)\Delta V_j \tag{7}$$

Here, the continuous space is partitioned into $N_s$ discrete regions (e.g., voxels), each with a volume $\Delta V_j$ centered at position $\mathbf{r}_j$. To computationally represent these regions, we introduce spatial nodes in addition to original atomic nodes. Each spatial node is placed at a specific coordinate $\mathbf{r}_j$ and represents the corresponding spatial region.

### 3.3.1 Region-wise Heterogeneous Message Passing

To model the interactions between atoms and these new spatial regions (i.e. $g(\mathbf{r})$), we convert the original isomorphic atomic graph into a heterogeneous graph composed of both atomic nodes and spatial nodes. We denote the spatial node and its feature vector as $s_j$ and $h_{s_j}$. The formulation of our *Region-wise Heterogeneous Message Passing* is as follows:

$$h_{s_j}^{t+1} = \mathcal{U}'_t(h_{s_j}^t, \{h_{v_i}^t, e_{v_i,s_j}\}_{v_i \in \mathcal{N}(s_j)}[, \{h_{s_k}^t, e_{s_k,s_j}\}_{s_k \in \mathcal{N}(s_j)}]) \tag{8}$$

$$h_{v_i}^{t+1} = \mathcal{U}_t(h_{v_i}^t, \{h_{v_j}^t, e_{v_i,v_j}\}_{v_j \in \mathcal{N}(v_i)}) \tag{9}$$

Here, $\mathcal{U}'_t$ and $\mathcal{U}_t$ are the update function for spatial and atomic nodes. The new feature vector $h_{s_j}^{t+1}$ aggregates messages not only from nearby atoms ($v_i \in \mathcal{N}(s_j)$) but also from adjacent spatial nodes ($s_k \in \mathcal{N}(s_j)$). In practice, the method for constructing spatial node adjacency nodes is ex-

actly the same as that for atoms, which is based on cutoff and the maximum number of neighbors. This inter-spatial node message passing is optional and depends on the physical nature of the target property. For example, when predicting gas adsorption capacity, message passing between spatial nodes is physically meaningful as it captures the cooperative interactions between guest molecules in adjacent regions. In contrast, for a property like accessible volume, where adjacent regions are independent, such interactions have no physical meaning and can be omitted. In this work, because the global receptive field of the adopted multi-modal Transformer method already enables interaction between spatial nodes (see Sec. 3.3.2), we ignore the message passing process between spatial nodes.

### 3.3.2 PROPERTY-ADAPTIVE READOUT VIA MULTI-MODAL ATTENTION

After obtaining atom and spatial node representations, one may directly pool the spatial-node features via

$$p = \sum_{j=1}^{N_s} \text{MLP}(h_{s_j}), \tag{10}$$

which already improves performance on spatial properties (Sec. 4). However, spatial nodes are specifically designed for spatially decomposable properties, whereas many widely used material properties remain atom-decomposable. Using spatial pooling alone introduces a mismatched inductive bias and degrades performance on these non-spatial tasks. Thus, an effective readout must *adaptively* combine atomic and spatial information.

To achieve this, we impose an ordering on spatial nodes with positional embeddings and feed the ordered spatial features together with unordered atomic features into a Transformer decoder (Vaswani et al., 2017). The attention-based decoder selectively integrates the two types of representations, enabling SpatialRead to retain the gains on spatial properties *without sacrificing* performance on conventional non-spatial tasks. A detailed architecture is provided in App. A.2.

## 4 EXPERIMENTS

Our experiments are designed to address the following three questions: (1) Does SpatialRead outperform node-decomposable methods on spatial properties? (2) What types of material features benefit most from SpatialRead? (3) For non-spatial properties, does the spatial inductive bias introduce any degradation?

**Datasets.** We first construct a dataset of spatial properties to evaluate the effectiveness of SpatialRead. The dataset mainly consists of four material types: (1) Metal Organic Frameworks (MOFs), (2) Covalent Organic Frameworks (COFs) (3) Porous Polymer Networks (PPNs), (4) zeolites, and two types of properties: (1) Geometric features such as topology type and pore-limiting diameter, computed using zeo++ (Willems et al., 2012). (2) Gas-related properties such as adsorption capacity and separation ratio, simulated through molecular dynamics. These materials are specifically selected because most of their properties, including adsorption capacity and separation ratio, are inherently spatial. Detailed descriptions of the dataset are provided in App. A.3.

**Baselines.** For spatial property prediction, we compare against the following baselines: (1) CGCNN (Xie & Grossman, 2018), a widely used MPNN designed for crystals. (2) Gem-Net (Gasteiger et al., 2021), an advanced invariant graph neural network that incorporates angular information, substantially improving performance over other MPNNs. (3) MOFormer (Cao et al., 2023), a contrastive pre-trained variant of CGCNN using SMILES representations. (4) MOF-Transformer (Kang et al., 2023), a multimodal framework with large-scale pre-training tailored for MOFs. We use the improved weights from PMTransformer (Park et al., 2023), which extends the pre-training dataset and enhances performance. (5) JMP (Shoghi et al., 2024), a GemNet-based foundation model pre-trained on 120 million molecular and material samples. Training details are provided in App. A.4. Another kind of baselines are different readout functions. However, as in most case for material / molecule property prediction (Schütt et al., 2017; Xie & Grossman, 2018; Gasteiger et al., 2020; 2021; 2022; Shoghi et al., 2024; Wang et al., 2024), modern GNNs in this field focus on the design of complex message passing process, while adopting the simple sum or mean pooling. Nevertheless, in order to ensure the completeness of the work, here we compare

SpatialRead with two most typical readout function GraphTrans (Wu et al., 2021) and GMT (Baek et al., 2021).

**Ablations.** In the following experiments, we consider three types of ablation settings. (1) Backbone GNN with simple sum / mean pooling of atom nodes, i.e. Base GNN. (2) Backbone GNN enhanced with Spatial Nodes, the graph feature will be pooled from all spatial nodes, i.e. Base GNN + SN (Spatial Node) (3) Backbone GNN processes heterogeneous graph of atom node and spatial node. A Multi-Modal Transformer architecture is used to process both atom features and spatial node features, i.e. Base GNN + SN + MM (Multi Modal).

Table 1: Performance (R2 Score) of SpatialRead on spatial properties in integral form

| | Model | MOF $C_3H_6/C_3H_8$ sep. | MOF $N_2$ ads. | MOF $CH_4/N_2$ sep. | COF $CH_4$ ads. | PPN $CH_4$ ads. | zeolite $CH_4$ heat. |
|---|---|---|---|---|---|---|---|
| Scratch | CGCNN | 0.663 | 0.760 | 0.718 | 0.556 | 0.692 | 0.411 |
| | GemNet (JMP from scratch) | 0.729 | 0.968 | 0.924 | 0.816 | 0.932 | 0.836 |
| | GemNet + SN + MM | 0.753 | 0.979 | 0.921 | 0.986 | 0.923 | 0.881 |
| Pretrain | MOFormer | 0.616 | 0.754 | 0.698 | 0.541 | 0.636 | 0.388 |
| | MOFTransformer | **0.817** | 0.918 | 0.905 | 0.967 | 0.942 | 0.836 |
| | JMP | 0.774 | 0.971 | 0.908 | 0.884 | 0.947 | 0.874 |
| | JMP + SN + MM | 0.792 | **0.988** | **0.941** | 0.982 | 0.969 | 0.945 |
| Scratch | PaiNN | 0.691 | 0.925 | 0.867 | 0.736 | 0.856 | 0.791 |
| | PaiNN + GraphTrans | 0.712 | 0.912 | 0.870 | 0.750 | 0.847 | 0.801 |
| | PaiNN + GMT | 0.740 | 0.924 | 0.866 | 0.742 | 0.863 | 0.803 |
| | PaiNN + SN (ours) | 0.794 | 0.978 | 0.936 | 0.979 | **0.978** | 0.886 |
| | PaiNN + SN + MM (ours) | 0.784 | 0.987 | **0.941** | **0.987** | 0.977 | **0.969** |

## 4.1 SPATIAL PROPERTIES

We first evaluate SpatialRead on six representative spatial properties, including: (1) gas adsorption in MOFs, (2) two gas separation tasks in MOFs, (3) gas adsorption in COFs, (4) gas adsorption in PPNs, and (5) adsorption heat in zeolites. Results are summarized in Table 1. Among the three representative MPNNs (CGCNN, GemNet, PaiNN), the ranking GemNet > PaiNN > CGCNN highlights the critical role of backbone design. Using complex readout functions including GraphTrans and GMT marginally improves the performance. Nonetheless, all pure MPNN approaches underperform MOFTransformer. While PaiNN lags behind MOFTransformer and JMP, augmenting it with spatial nodes (+SN) yields notable improvements across tasks. Adding the attention mechanism (MultiModal, +MM) does not lead to further consistent gains. This can be attributed to the fact that these properties can naturally be expressed as the sum of regional contributions. Hence, equation 10 is already physically grounded, and introducing additional global receptive fields has limited physical justification. This stands in contrast to the results in Sec. 4.3, where global receptive fields are indispensable. Overall, these results highlight the effectiveness of spatial nodes in modeling spatial properties: by decomposing properties into regional contributions, performance can be substantially enhanced without altering the backbone architecture.

## 4.2 BENEFIT FROM ALREADY-PRE-TRAINED FOUNDATION MODEL

Despite being designed for general GNNs such as PaiNN, it is also important to assess whether SpatialRead can be directly applied to a pre-trained foundation model like JMP (Shoghi et al., 2024). Surprisingly, although JMP is pre-trained with a simple sum-pooling readout, it can be seamlessly enhanced by SpatialRead. As shown in Table 1, both GemNet (i.e., JMP trained from scratch) and JMP experience substantial performance gains when equipped with SpatialRead, and JMP+SpatialRead further outperforms GemNet+SpatialRead, demonstrating that large-scale pre-training remains beneficial even when the downstream readout differs from that used during pre-training. These results highlight the versatility of SpatialRead with complex multi-body and pre-trained GNNs. We note that JMP+SpatialRead does not always surpass PaiNN+SpatialRead, likely due to the higher computational cost of GemNet's four-body interactions and the resulted limited hyperparameter settings, as discussed in Appendix A.5.

Table 2: Performance (R2 Score) of SpatialRead on geometric properties

|  | **Model** | ASA | VF | PLD | LCD |
|---|---|---|---|---|---|
| Scratch | CGCNN | 0.984 | 0.883 | 0.536 | 0.565 |
|  | GemNet | 0.994 | 0.977 | 0.586 | 0.667 |
| Pretrain | MOFormer | 0.979 | 0.894 | 0.563 | 0.624 |
|  | MOFTransformer | 0.916 | 0.989 | **0.966** | 0.970 |
|  | JMP | 0.995 | 0.985 | 0.585 | 0.650 |
| Scratch | PaiNN | 0.993 | 0.951 | 0.594 | 0.631 |
|  | PaiNN + SN (ours) | 0.974 | **0.999** | 0.856 | 0.913 |
|  | PaiNN + SN + MM (ours) | **0.996** | **0.999** | 0.965 | **0.975** |

## 4.3 GLOBAL GEOMETRIC PROPERTIES

In the previous section, we demonstrated that for spatial properties that can be naturally expressed in integral form, SpatialRead provides significant improvements by reformulating the readout as a summation over spatial regions. However, not all spatial properties admit such a simple integral formulation. To further examine this distinction, we evaluate SpatialRead on several representative geometric properties, as reported in Table 2. For accessible surface area (ASA), the property is essentially node-decomposable: surface area is determined primarily by the atoms located at the boundary. In this case, directly introducing spatial nodes provides negative influence. Nevertheless, since our multimodal architecture adaptively balances atom- and space-decomposable inductive biases, SpatialRead still achieves a slight improvement over the pure PaiNN baseline. Void fraction (VF) can be expressed in integral form, but the task itself is relatively simple (Kang et al., 2023), and thus most models already achieve high performance. Here, SpatialRead again yields marginal improvements, confirming its robustness without incurring degradation. The situation is markedly different for pore-limiting diameter (PLD) and largest cavity diameter (LCD). These descriptors cannot be represented as integrals over local contributions, but are instead better understood as functionals of the signed distance function (SDF) of the material geometry. Their values depend on the global shape of the pore space rather than additive regional properties. In such cases, equation 10 is no longer physically meaningful. The introduction of spatial nodes already provides significant gains, while the Transformer-based multimodal architecture further enhances performance through its global receptive field and expressive capacity. These results suggest that SpatialRead is effective not only for integral-type spatial properties, but also for more complex forms.

Table 3: Performance of SpatialRead on MatBench

| **Task** | **MODNet** | **coGN** | **JMP** | **JMP + SpatialRead** |
|---|---|---|---|---|
| JDFT2D (meV/atom) | 25.55 | 22.25 | 20.72 | **18.17** |
| Phonons ($cm^{-1}$) | 34.77 | 32.12 | 26.6 | **25.8** |
| Dielectric (unitless) | 0.169 | 0.178 | **0.133** | **0.133** |
| Log GVRH (log10(GPa)) | 0.073 | 0.068 | **0.06** | **0.06** |
| Log KVRH (log10(GPa)) | 0.054 | 0.052 | **0.044** | 0.047 |
| Perovskites (eV/unitcell) | 0.093 | **0.027** | 0.029 | 0.030 |
| MP Gap (eV) | 0.215 | 0.153 | 0.119 | **0.107** |
| MP Formation Energy (meV/atom) | 40.2 | 17.4 | **13.6** | 15.3 |

## 4.4 MATBENCH RESULTS FOR OTHER COMMON MATERIAL PROPERTIES

While the primary strength of SpatialRead lies in modeling spatial properties, it is also important to ensure that it does not degrade performance on standard material-property benchmarks. As shown in Table 3, the foundation model JMP already surpasses leading baselines on the MatBench dataset. Adding SpatialRead maintains comparable performance across most tasks. Two tasks behave differently. For MP–E–Form, SpatialRead slightly reduces accuracy, whereas for bandgap, SpatialRead leads to an improvement. This difference is consistent with the inherent nature of the two quantities. Formation energy is an atom-decomposable property, making a simple summation pooling scheme,

such as the one used in JMP, a suitable inductive bias. In contrast, the bandgap is defined as the energy difference between the valence band maximum (VBM) and the conduction band minimum (CBM). These frontier electronic states are formed by collective contributions from multiple atoms and cannot be meaningfully represented as a sum or average over atom-wise descriptors. Therefore, simple atom pooling provides an inappropriate inductive bias for bandgap prediction. An attention-based readout, which allows the model to selectively weight atoms according to their relevance to the VBM and CBM, is naturally more suitable. This improvement is attributable to the attention mechanism rather than the presence of spatial nodes. These results further reflect the importance of designing physically-grounded readout function. In summary, SpatialRead preserves performance on non-spatial tasks even when combined with a strong pre-trained model, confirming its versatility across diverse material-property settings.

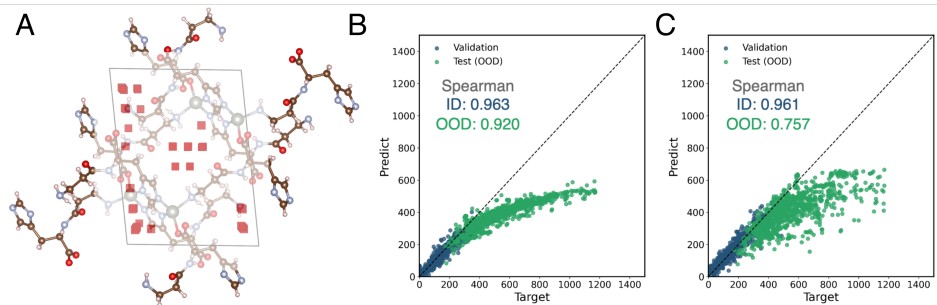

Figure 3: **Working mechanism of Spatial Nodes. (A) Visualization of spatial nodes with high contribution.** Those spatial nodes with top 10% contributions are drawn as red cubes. The drawn important spatial nodes are mainly located in the pore regions of the material. **(B) Out of distribution generalization for PaiNN+SumPooling. (C) Out of distribution generalization for PaiNN+SpatialRead.** The adsorption capacity data is separated according to the void fraction. We choose 1/7 materials with the most high void fraction as the test set to test the out-of-distribution generalization ability of different readout function. PaiNN+SpatialRead is trained for 3 epochs while PaiNN+SumPooling is trained for 40 epochs to make sure their precision on in-distribution data is similar. PaiNN + SpatialRead maintains better spearman correlation coefficient in out-of-distribution data.

## 4.5 INTERPRETATION OF SPATIAL NODES

We now provide an interpretation of how spatial nodes function within SpatialRead. In summary, those sparse regions with few or no atoms are hard to be described by conventional atom-decomposable methods. However, these regions are critical for some spatial properties. Thus, when a material or graph is highly non-uniform, one should consider adding spatial nodes to those sparse regions. To illustrate this, we employ SchNet (Schütt et al., 2017) as a simple backbone with an MLP readout (equation 10), trained to predict gas adsorption capacity. In this setting, the prediction is obtained by averaging node outputs, allowing each node's output to be interpreted as its *contribution*. We visualize the top 10% most contributing spatial nodes in Fig. **??**. These nodes are predominantly located within the pore channels of MOFs, highlighting that SpatialRead effectively identifies the structural regions most relevant to adsorption. Under a purely node-decomposable inductive bias, such pore-level contributions will be implicitly learned to be assigned to the nearby atoms, which may increase the difficulty of learning on the network and its generalization ability. In addition to the case study, we also conducted statistical analysis in the Appendix A.8.

## 4.6 PHYSICAL-GROUNDED INDUCTIVE BIAS FOR BETTER GENERALIZATION

A natural way to examine whether a model has learned physically meaningful features is to test its behavior under distribution shift. For a spatial property, although a typical GNN can implicitly infer critical regions from atomic environments, this atom-based decomposition becomes unstable when faced with distribution shift. To validate this, we construct an out-of-distribution (OOD) split by placing the highest-porosity 1/7 MOFs entirely into the test set. To ensure fairness, we early-stop PaiNN+SpatialRead such that its in-distribution accuracy matches that of the base PaiNN.

Details of the calculation method, data split, and validation metric can be found in A.7. While both models experience increased error due to the severe shift in porosity, PaiNN+SpatialRead maintains significantly higher ranking stability (Spearman $0.92/0.95$ vs. $0.76$). Fig. **??** also demonstrates that the prediction variance of PaiNN+SpatialRead is smaller than PaiNN+SumPooling, indicating the robust prediction. This ability is of crucial importance for the screening of new materials. Because for high-throughput screening, people are more concerned about whether the performance ranking of different materials is correct rather than the actual prediction error. In summary, the benefit of SpatialRead lies in guiding the model toward physically plausible solutions within this function class, thereby improving robustness of prediction, especially under distribution shift.

## 4.7 SAMPLING STRATEGY AND COMPUTATIONAL COST

Since the position of the introduction of spatial nodes is not unique. The sampling strategy will significantly affect the model performance. Besides the sampling strategy based on grid/fractional coordinates, another method is to sample based on resolution. The resolution-based method allocates different numbers of spatial nodes for different-sized cells. Resolution-based sampling results in better performance on large system like COFs. Detailed experiment can be found in Appendix A.9. Due to space limitations, computational cost are discussed in Appendix A.5 and A.10. For a typical MOF material containing about 300 atoms, adding SpatialRead adds about 30% computational burden. Increasing the number of spatial nodes consistently improve the performance. $8 * 8 * 8$ points provide a good balance between computational burden and performance.

## 5 CONCLUSION

In this work, we revisited the inductive bias of MPNN readouts, noting that the common assumption of node-decomposability is insufficient for many spatial properties. To address this, we proposed SpatialRead, which augments the atomic graph with spatial nodes and employs a multimodal Transformer to adaptively select between atomic- and spatial-decomposable representations. Extensive experiments show that SpatialRead substantially improves predictions of spatial properties such as gas adsorption capacity, pore limiting diameter (PLD), etc, while remaining the performance of the backbone MPNN in other non-spatial properties. Contribution analysis demonstrates that spatial nodes naturally capture critical regions. These benefits incur only modest computational overhead, establishing SpatialRead as a practical, scalable framework that incorporates spatial inductive bias into graph neural networks. SpatialRead emphasizes the importance of designing physically-grounded readout function for the target property, which is commonly ignored in current MPNNs.

### AUTHOR CONTRIBUTIONS

J.Z. proposes the main idea, conducts the experiments, and writes the paper. Z.W. and H.Q. help with the experiments. W.T. and B.Y. supervise the progress of the project and provide critical feedback on the paper.

### ACKNOWLEDGMENTS

We gratefully acknowledge support for this work provided by National Natural Science Foundation of China (NSFC) (Grant Nos.: 623B1004 (J.Z.), 62472102 (B.Y.)), National Key R&D Program of China (2024YFC3406500 and 2024YFC3406501), the Natural Science Foundation of Shanghai (24ZR1490400 to W. T.).

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

# A  APPENDIX

## A.1  PROOFS

### A.1.1  PROOF OF THE EQUIVALENCE BETWEEN LOCAL DESCRIPTION FUNCTION AND CONTRIBUTION FUNCTION

Consider a graph $G = (V, E)$ with node set $V$. The target property $p$ is assumed to be a function of the nodes, i.e. $p = p(V)$. A message-passing graph neural network (MPNN) can be regarded as consisting of a *local description function c* and a *readout function f*. Each node can only perceive the information within its receptive field $r_{mp}$. Formally,

$$h_{v_i} = c(\{v_j \mid v_j \in V, \, r(v_i, v_j) < r_{mp}\}) \tag{11}$$

$$H = \{h_{v_i} \mid v_i \in V\} \tag{12}$$

$$h_{\text{graph}} = f(H) \tag{13}$$

Without loss of generality, the readout function can always be written in a node-decomposable form:

$$h_{\text{graph}} = f(H) = \sum_{v_i \in V} f(h_{v_i} \mid H) \tag{14}$$

This is because when $f$ has an unlimited receptive field, its output can at least be evenly distributed to each node. In practice, for most target properties (such as total energy and most properties in QM9 except dipole moment), $f$ is implemented as an MLP depending only on $h_{v_i}$. More generally, when $f$ has a finite receptive field $r_{read}$, there always exists an equivalent local description function $c'$ defined on an expanded neighborhood $\mathcal{N}(\mathbf{pos}_i) = \{v_j \mid r(v_i, v_j) < \max(r_{mp}, r_{read})\}$ such that

$$h'_{v_i} = f(h_{v_i} \mid H) = c'(\mathcal{N}(\mathbf{pos}_i)) \tag{15}$$

$$h_{\text{graph}} = \sum_{v_i \in V} h'_{v_i} \tag{16}$$

**Region-decomposable formulation.**  We now reformulate the graph-level feature as an integral over space:

$$h_{\text{graph}} = \int g(\mathbf{r}) \, d^3\mathbf{r} \tag{17}$$

$$g(\mathbf{r}) = g(\mathcal{N}(\mathbf{r})) \qquad \text{where } \mathcal{N}(\mathbf{r}) = \{v_j \mid \|\mathbf{r} - \mathbf{pos}_j\| < r_g\} \tag{18}$$

We will show that equation 16 and equation 17 are *equivalent in expressive power*, by constructively defining a mapping from one form to the other and vice versa.

**From node-decomposable to region-decomposable.**  Define $g$ using Dirac delta functions:

$$g(\mathbf{r}) = \sum_{v_i \in V, v_i \in \mathcal{N}(\mathbf{r})} h'_{v_i} \, \delta(\mathbf{r} - \mathbf{pos}_i) \tag{19}$$

Since $\int \delta(\mathbf{r} - \mathbf{pos}_i) d^3\mathbf{r} = 1$, we have

$$\int g(\mathbf{r}) \, d^3\mathbf{r} = \sum_{v_i \in V} h'_{v_i} \int \delta(\mathbf{r} - \mathbf{pos}_i) d^3\mathbf{r}$$

$$= \sum_{v_i \in V} h'_{v_i} = h_{\text{graph}} \tag{20}$$

Thus any node-decomposable form can be expressed as a region-decomposable form.

**From region-decomposable to node-decomposable.**  Assume $g(\mathbf{r})$ has a finite receptive field $r_g$. For each position $\mathbf{r}$, define a normalized weight over nearby nodes:

$$\hat{w}_i(\mathbf{r}) = \begin{cases} \frac{1}{\|\mathbf{r} - \mathbf{pos}_i\|}, & \|\mathbf{r} - \mathbf{pos}_i\| < r_g \\ 0, & \text{otherwise} \end{cases} \tag{21}$$

$$w_i(\mathbf{r}) = \frac{\hat{w}_i(\mathbf{r})}{\sum_{v_j \in V, v_j \in \mathcal{N}(\mathbf{r})} \hat{w}_j(\mathbf{r})} \tag{22}$$

For any $\mathbf{r}$ we have $\sum_{v_i \in V} w_i(\mathbf{r}) = 1$. Therefore,

$$\int g(\mathbf{r})\, d^3\mathbf{r} = \int g(\mathbf{r}) \sum_{v_i \in V} w_i(\mathbf{r})\, d^3\mathbf{r}$$

$$= \sum_{v_i \in V} \int g(\mathbf{r}) w_i(\mathbf{r})\, d^3\mathbf{r}$$

$$= \sum_{v_i \in V} \int_{\|\mathbf{r}-\mathbf{pos}_i\| < r_g} g(\mathbf{r}) w_i(\mathbf{r})\, d^3\mathbf{r} \tag{23}$$

We can thus define a local description function for each node as

$$c(v_i) = \int_{\|\mathbf{r}-\mathbf{pos}_i\| < r_g} g(\mathbf{r}) w_i(\mathbf{r})\, d^3\mathbf{r} \tag{24}$$

This gives a node-decomposable representation that is equivalent to the original region-decomposable one.

**Remarks.** Note that the mapping from region to node is not unique, because the choice of weights $w_i(\mathbf{r})$ is arbitrary. This implies that *inductive bias is crucial* in practice. For example, when predicting total energy, because node-level labels are unavailable, one could (in principle) assign all the energy of a methyl group to its carbon atom and zero to its hydrogens. Although this rule is learnable, it is physically incorrect. Therefore, additional inductive biases such as bond length, bond angle, or dihedral angle are typically introduced to guide the model.

**Conclusion.** In summary, any node-decomposable representation can be mapped to a region-decomposable one, and vice versa. Hence, these two formulations are *expressively equivalent*: they do not change the representational power of GNNs, but merely reflect different inductive biases.

### A.1.2 PROOF OF THE GLOBAL RECEPTIVE FIELD

**On realizing a global readout with local encoders.** Even when the readout function $f$ possesses a *global* receptive field, the local description function $c$ can still be restricted to have only a *local* receptive field. This can be achieved by the following construction.

For each node $v_i$, let its local descriptor be

$$h_{v_i} = c(v_i), \tag{25}$$

which depends only on a bounded neighborhood of $v_i$. Assign $h_{v_i}$ to a small spatial region $R_i$ surrounding the node position $\mathbf{pos}_i$, and define the spatial field

$$g(\mathbf{r}) = \sum_{v_i \in V} h_{v_i}\, \chi_{R_i}(\mathbf{r}), \tag{26}$$

where $\chi_{R_i}(\mathbf{r})$ is the indicator function of region $R_i$.

Because the number of nodes is finite, one can always choose regions $R_i$ that are sufficiently small and pairwise disjoint. Let the volume of each region be $V_i = |R_i|$. Then, within $R_i$ the field is constant:

$$g(\mathbf{r}) = h_{v_i}, \qquad \mathbf{r} \in R_i. \tag{27}$$

The field $g(\mathbf{r})$ can be discretized as a finite-resolution 3D image by sampling on a grid with voxel size $\delta V$ such that each $R_i$ occupies at least one voxel. This yields a tensor representation $\{g(\mathbf{r}_j)\}_{j=1}^{N_s}$ of finite spatial resolution. Although this representation cannot capture ideal Dirac delta functions (which would require infinite resolution), it can losslessly represent the piecewise-constant field constructed above because each $R_i$ is non-overlapping.

Finally, the global readout can be realized as a general function operating on this spatial field:

$$p = f\big(\{h_{v_i}\}_{i \in V}\big) \;\equiv\; F\big(\{g(\mathbf{r}_j)\}_{j=1}^{N_s}\big), \tag{28}$$

where $F$ can be any architecture with a global receptive field (e.g. a Transformer or a CNN operating on the 3D grid). In this way, the local descriptors $c(v_i)$ remain strictly local, while the global dependency is handled solely by the subsequent global network $F$. This shows that even if the target readout $f$ is global, it can be implemented by composing a local encoder $c$ (on the graph) with a global readout $F$ (on the discretized spatial field), without violating the locality constraint on $c$.

A.2 DETAILED ARCHITECTURE OF SPATIALREAD

Given a material $G = (V, E)$ with atomic positions $\{\mathbf{r}_i\}$, atomic types $\{x_i\}$, and lattice vectors $\mathbf{L}$, SpatialRead operates as follows.

We place spatial nodes on a uniform $M \times M \times M$ grid inside the unit cell, resulting in a total of $M^3$ spatial nodes. The fractional coordinates of the spatial node indexed by $(i, j, k)$ are given by

$$\mathbf{r}_s(i, j, k) = \left( \frac{i}{M}, \frac{j}{M}, \frac{k}{M} \right), \qquad i, j, k = 0, \dots, M - 1. \tag{29}$$

Next, we transform the fractional coordinates to cartesian coordinates according to the lattice vector. This uniform sampling scheme provides full coverage of the 3D domain with a spatial resolution controlled by $M$.

Next, we construct a heterogeneous graph composed of two types of nodes: atomic nodes ($v_i \in V$) and spatial nodes ($s_j$). Edges are built based on Euclidean distance with a cutoff radius $r_{\text{cut}}$, typically set to 5–8 Å, and respecting a maximum neighbor limit for efficiency. Crucially, we allow two types of edges:

- **Atom–atom edges:** $(v_i, v_j)$ if $\|\mathbf{r}_i - \mathbf{r}_j\| \leq r_{\text{cut}}$
- **Atom–spatial edges:** $(v_i, s_j)$ if $\|\mathbf{r}_i - \mathbf{r}_{s_j}\| \leq r_{\text{cut}}$

No edges are allowed from spatial nodes to atomic nodes, enforcing unidirectional information flow: *atoms influence space, but not vice versa.* Notably, periodic boundary conditions were taken into account, as the materials being dealt with in this work are all crystals.

Message passing is performed using a PaiNN-style MPNN (Schütt et al., 2021), which jointly updates scalar and vector node features through interactions along edges. The process runs for $T$ layers, updating atomic and spatial node features, without altering the original MPNN backbone. At the final layer, we obtain:

- An unordered set of atomic feature vectors: $\{\mathbf{h}_{v_i}^T\}_{i=1}^{|V|}$
- An ordered list of spatial node feature vectors: $\{\mathbf{h}_{s_j}^T\}_{j=1}^{512}$

To process these heterogeneous features, we adopt a multi-modal Transformer decoder (Vaswani et al., 2017). Since the number of spatial nodes is fixed (i.e., $M^3$), and their order is determined by the $i, j, k$ index of each node in equation 29, we can train a learnable position embedding $p$ of shape $[M^3, F]$. We add the learnable positional embedding to each spatial node feature:

$$\widetilde{\mathbf{h}}_{s_j} = \mathbf{h}_{s_j}^T + \mathbf{p}_j, \tag{30}$$

where $\mathbf{p}_j$ is a learnable embedding encoding the 3D index of the voxel. A [CLS] token with a learned initial embedding $\mathbf{h}_{[\text{CLS}]}$ is prepended to the sequence of spatial node features.

The atomic features are padded to a fixed maximum length $M$ (e.g., $M = 200$) to handle variable-sized crystals. The full input to the Transformer decoder is:

$$\text{Input} = \left[ \mathbf{h}_{[\text{CLS}]}, \widetilde{\mathbf{h}}_{s_1}, \dots, \widetilde{\mathbf{h}}_{s_{512}} \right], \left[ \mathbf{h}_{v_1}, \dots, \mathbf{h}_{v_{|V|}}, \mathbf{0}, \dots, \mathbf{0} \right] \tag{31}$$

The Transformer decoder, consisting of $L$ attention layers, processes this sequence via self-attention and cross-attention mechanisms. Importantly, no causal masking is applied, allowing full interaction among all tokens. After processing, the final state of the [CLS] token is used for prediction:

$$\mathbf{h}_{[\text{CLS}]}^{\text{out}} = \text{TransformerDecoder}(\text{Input}), \quad p = \text{MLP}(\mathbf{h}_{[\text{CLS}]}^{\text{out}}) \tag{32}$$

where $p$ is the predicted property.

In the ablation setting `spnode_mlp`, we bypass the Transformer and instead use a simpler readout: each spatial node feature is processed independently by an MLP to yield a scalar output $o_j = \text{MLP}(\mathbf{h}_{s_j}^T)$, and the final prediction is the average:

$$p = \frac{1}{N_s} \sum_{j=1}^{N_s} o_j \tag{33}$$

When spatial nodes are absent, this reduces to averaging atomic outputs—equivalent to conventional numeric-level pooling ( equation 3).

Table 4: Hyperparameters of the PaiNN backbone and Transformer decoder used in SpatialRead.

| Component | Parameter | Value |
|---|---|---|
| PaiNN | Number of layers | 6 |
| | Hidden dimension | 128 |
| | Filter dimension | 128 |
| | Cutoff radius $r_{\text{cut}}$ | 6.0 Å |
| | Maximum neighbors | 30 |
| Transformer Decoder | Number of layers | 6 |
| | Hidden dimension | 128 |
| | Number of attention heads | 8 |
| | Feed-forward dimension | 512 |
| | dropout | 0.15 |
| | Maximum atomic count (padding) | 1024 |
| Spatial Nodes | Number of spatial nodes $N_s$ | 512 ($8^3$) |
| | Position embedding size | 128 |

Table 5: Training strategy and optimization hyperparameters used in SpatialRead.

| Training Parameter | Value |
|---|---|
| Total epochs | 80 |
| Optimizer | AdamW |
| Learning rate (initial) | $1 \times 10^{-4}$ |
| Learning rate scheduler | ReduceLROnPlateau |
| Monitor metric | `val_loss` |
| Mode | `min` |
| Patience | 10 |
| Factor | 0.8 |
| Threshold | $1 \times 10^{-4}$ |
| Minimum learning rate | $1 \times 10^{-6}$ |
| Weight decay | 0.0 |
| Batch size | 8 |

## A.3 DATASETS

The dataset mainly includes four material types:

- Metal Organic Frameworks (MOFs): 23157 samples from Chung et al. (2019), Tang et al. (2021), Gulbalkan et al. (2023), and Kang et al. (2023).
- Covalent Organic Frameworks (COFs): 7000 samples from Hu et al. (2015) and Deeg et al. (2020).
- Porous Polymer Networks (PPNs): 7000 samples from Martin et al. (2014).
- zeolites: 7000 samples from Kim et al. (2020).

Task types include:

- Topology Type
- Void Fraction
- Accessible Surface Area
- Pore Limited Diameter

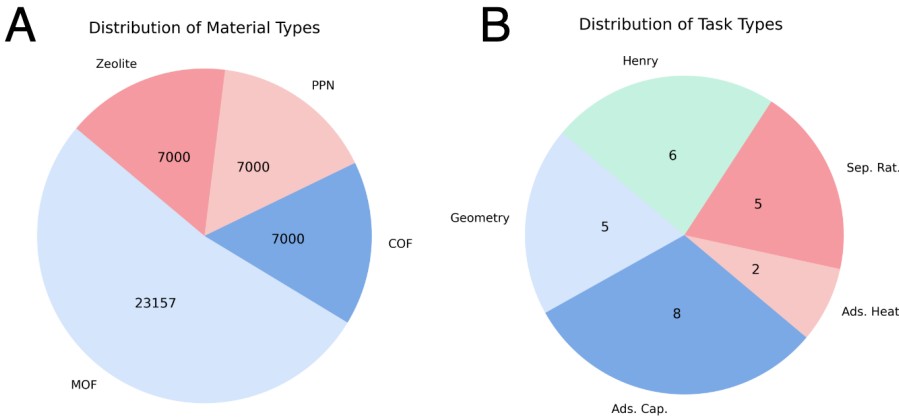

Figure 4: **Dataset of spatial properties for porous materials. (A) Distribution of material types.** The dataset contains four types of porous materials: (1) Metal Organic Frameworks, (2) Covalent Organic Frameworks, (3) Porous Polymer Networks, (4) zeolites. **(B) Distribution of task types.** Tasks mainly include five types: (1) Geometric Features, such as void fraction, accessible surface area etc. (2) Gas adsorption capacity, (3) Separation ratio, (4) adsorption heat, (5) Henry's constant.

- Largest Cavity Diameter

The geometric features are calculated by Willems et al. (2012). The probe radius is set to 0.5. The number of sampling points are set to 2,000 for surface area and 50,000 for volume. These datasets are randomly split into train, validation and test set according to 5 : 1 : 1. All material structures can be found in Bobbitt et al. (2023), which are collected from Wilmer et al. (2012) and Chung et al. (2019).

The source of the dataset, as well as the sizes of the training, validation and test sets, are shown in Table 6.

### A.4   TRAINING DETAILS OF BASELINE MODELS

**CGCNN** (Xie & Grossman, 2018): CGCNN is one of the most typical MPNN for crystal. But as a kind of graph convolutional neural network, the performance has lagged behind modern methods. Despite, it still reflect the basic feature of Graph Convolutional Network. The model architecture follows the original CGCNN. We train CGCNN for 300 epochs, the learning rate is set to 0.01 and decays to 0.001 and 0.0001 in 100 and 200 epochs. Other settings follow the original CGCNN.

**MOFormer** (Cao et al., 2023): MOFormer is a kind of pre-training method, which uses contrastive learning in material structure and SMILES code. MOFormer pretrain CGCNN in about 300,000 MOFs. To achieve full training, we fine-tune the pre-trained CGCNN for 60 epochs instead of the default 30 epochs.

**JMP** (Shoghi et al., 2024): JMP is a GemNet (Gasteiger et al., 2021) pre-trained on 120 million material and molecule. We fine-tune JMP for 80 epochs. All training strategy and model architecture follows the default setting. Nevertheless, the default setting of JMP on qMOF Rosen et al. (2021; 2022) dataset is not suitable for our task. Most MOFs in the qMOF dataset is smaller than CoREMOF. In original settings, JMP adopts an adaptive strategy to set cutoff and max_num_neighbors. The cutoff is fixed to 19.0 Å. But when the number of atoms is larger than 300, the max_num_neighbors is set to 5, which is too small. A large number of MOFs in the CoRE-MOF (where most structures are real and obtained by experiments) dataset is larger than 300. We fixed the max_num_neighbors to 8, which achieves balance between performance and computation cost.

**GemNet** (Gasteiger et al., 2021): Considering that JMP is a re-implementation of GemNet, we use the same code of JMP instead of the original implementation. We train the code of JMP without

Table 6: Details of the dataset

| Dataset | Source | Task | Unit | Training data | Val Data | Test Data |
|---|---|---|---|---|---|---|
| Geo | Kang et al. (2023) Willems et al. (2012) | Topology | - | 4,900 | 1,050 | 1,050 |
| | | VF | - | 4,900 | 1,050 | 1,050 |
| | | ASA | $m^2$/g | 4,900 | 1,050 | 1,050 |
| | | PLD | Å | 4,900 | 1,050 | 1,050 |
| | | LCD | Å | 4,900 | 1,050 | 1,050 |
| CoREMOF | Chung et al. (2019) | $N_2$ Ads. | $cm^3$(STP)/g | 5,000 | 1,000 | 1,000 |
| | | Ar Ads. | $cm^3$(STP)/g | 5,000 | 1,000 | 1,000 |
| CH4/N2 | Gulbalkan et al. (2023) | $CH_4$ Henry | mol/kg/Pa | 5,000 | 1,000 | 1,000 |
| | | $N_2$ Henry | mol/kg/Pa | 5,000 | 1,000 | 1,000 |
| | | $CH_4$/$N_2$ Sel. (0.1bar) | - | 5,000 | 1,000 | 1,000 |
| | | $CH_4$/$N_2$ Sel. (1bar) | - | 5,000 | 1,000 | 1,000 |
| | | $CH_4$/$N_2$ Sel. (10bar) | - | 5,000 | 1,000 | 1,000 |
| C3H6/C3H8 | Tang et al. (2021) | $C_3H_6$ Ads. | mol/kg | 1354 | 170 | 170 |
| | | $C_3H_8$ Ads. | mol/kg | 1354 | 170 | 170 |
| | | $C_3H_6$/$C_3H_8$ Sel. (1bar) | - | 1354 | 170 | 170 |
| | | $C_3H_6$/$C_3H_8$ Sel. (infinite) | - | 1354 | 170 | 170 |
| | | TSN_S (1bar) | - | 1354 | 170 | 170 |
| | | $C_3H_6$ Henry (298K) | log(mol/kg/Pa) | 1354 | 170 | 170 |
| | | $C_3H_8$ Henry (298K) | log(mol/kg/Pa) | 1354 | 170 | 170 |
| PPN | Martin et al. (2014) | $CH_4$ Ads. (65bar) | $cm^3$ (STP) / $cm^3$ | 5,000 | 1,000 | 1,000 |
| | | $CH_4$ Ads. (1bar) | $cm^3$ (STP) / $cm^3$ | 5,000 | 1,000 | 1,000 |
| COF | Mercado et al. (2018) Deeg et al. (2020) | $CH_4$ Ads. (65bar) | v (STP) / v | 5,000 | 1,000 | 1,000 |
| | | $CH_4$ Ads. (5.8bar) | v (STP) / v | 5,000 | 1,000 | 1,000 |
| | | $CO_2$ Ads. Heat. | kj/mol | 5,000 | 1,000 | 1,000 |
| | | $CO_2$ Henry | log(mol/kg/Pa) | 5,000 | 1,000 | 1,000 |
| zeolite | Kim et al. (2020) | $CH_4$ Henry | - | 5,000 | 1,000 | 1,000 |
| | | $CH_4$ Ads. Heat | kj/mol | 5,000 | 1,000 | 1,000 |

loading the pre-trained checkpoint to obtained the results of GemNet. Other strategies are the same as JMP.

**MOFTransformer/PMTransformer** (Kang et al., 2023; Park et al., 2023): MOFTransformer is a transformer-based multimodal network pre-trained on about 1 million MOFs. The subsequent work PMTransformer further uses 1.9 million porous materials as the pre-training dataset. Here we use the checkpoint of PMTransformer and the original implementation. The only difference is that the default setting finetune the model for 30 epochs, which is insufficient to converge. Thus we finetune the model for 60 epochs.

### A.5 COMPLEXITY OF SPATIAL NODES AND SCALABILITY TO LARGER SYSTEM

SpatialRead extends conventional MPNNs by introducing spatial nodes and a multimodal Transformer head, raising the question of computational overhead. Using SchNet (Schütt et al., 2017) as a backbone for controlled experiments, we find that increasing spatial node resolution improves performance until convergence around $8 \times 8 \times 8$ nodes, corresponding to roughly 1 Å$^3$ per node for typical porous materials. Based on this, we recommend 512 spatial nodes for materials such as MOFs, COFs, PPNs, and zeolites. Table 7 shows that adding spatial nodes increases training time by about 30% compared with the baseline PaiNN, while the full SpatialRead remains lightweight (2.9 MB, 2.78 min/epoch), with most overhead arising from message passing rather than the Transformer head. Overall, SpatialRead delivers substantial performance gains with only modest computational cost.

As demonstrated in A.2, the graph construction behavior of spatial nodes is similar to atoms. Each spatial node receive message from the neighboring atoms, which are determined by cutoff and maximum number of neighbors. The complexity of modern GNNs is commonly linearly related to the number of atoms. For example, for a GNN like PaiNN (Schütt et al., 2021) that takes into account the interaction between two bodies, its complexity is $O(Nk)$, where $N$ is the number of atoms and $k$ is the number of neighboring nodes. For a model like GemNet that takes into account interactions among up to four bodies, its complexity is approximately $O(Nk^3)$. Therefore, when the number of spatial nodes is fixed as a constant $M$, the model complexity is increased to $O(N + M)k$ or

Table 7: Computational cost in the CoREMOF dataset

| Model | Params (MB) | Training time / epoch (min) |
|---|---|---|
| JMP/GemNet | 38.5 | 3.77 |
| PaiNN | 1.3 | 2.01 |
| PaiNN + SN (ours) | 1.3 | 2.61 |
| PaiNN + SN + MM (ours) | 2.9 | 2.78 |

$O(N + M)k^3$. As the system becomes larger, the additional complexity brought about by the increase in spatial nodes can be almost negligible. However, it should be noted that a larger system typically implies a larger space, and therefore may require more spatial nodes to maintain a reliable resolution.

Due to the consideration of three-body and four-body interactions as well as higher embedding and edge feature encoding, the size of the memory usage and the training load increase rapidly with the number of neighbors. For actual MOFs like in the CoREMOF (Chung et al., 2019) dataset, the NVIDIA RTX GeForce 4090 (24 GB) only allows us to set the maximum value of max_num_neighbors to 15 (as a comparison, the original setting of JMP in MOFs set the max_num_neighbors to 5). Properties such as adsorption capacity are significantly influenced by the intermolecular interactions, and therefore may be more sensitive to parameters like cutoff and max_num_neighbors. Even under such a disadvantage, JMP still achieved performance comparable to that of PaiNN. It can be expected that if the complete 30 maximum neighbors are enabled, the effect of JMP will surpass that of PaiNN. As for PaiNN, due to its simple two-body message passing process, we can allow each atom to have up to 30 neighboring nodes.

When attention-based modules are employed, the dominant cost arises from the $O(N^2)$ attention over atoms, rather than from the spatial nodes themselves. Since long-sequence attention is not the focus of this work, we evaluate the spatial-node overhead using the Spatial Node + MLP design, whose contribution remains size-independent. Specifically, we choose CoREMOF as the base dataset, since it is the most common dataset used for MOFs. We construct supercell to enlarge the system size and evaluate the computational cost of PaiNN (+ SpNode). We evaluate how long and how much memory are needed to train a PaiNN (with SpNode) in 5,000 MOFs in the NVIDIA GeForce RTX 4090.

The empirical results in Table 8 confirm that the runtime and memory overhead of spatial nodes remain effectively constant when the number of spatial nodes is fixed. As the system size grows, the relative impact of this overhead diminishes. Nevertheless, for sufficiently large systems, additional spatial nodes may be required to maintain spatial resolution, in which case the total cost may scale proportionally with the number of inserted spatial nodes.

Table 8: Training time and memory overhead introduced by spatial nodes at different system sizes.

| System Size | Model | Time / epoch (min) | Memory (MB) |
|---|---|---|---|
| 294 | PaiNN | 4.07 | 484 |
| 294 | PaiNN + SpNode | 5.21 | 546 |
| 634 | PaiNN | 4.86 | 752 |
| 634 | PaiNN + SpNode | 5.83 | 819 |
| 3092 | PaiNN | 10.7 | 2475 |
| 3092 | PaiNN + SpNode | 11.3 | 2520 |
| 8476 | PaiNN | 29.2 | 5857 |
| 8476 | PaiNN + SpNode | 26.4 | 5896 |

## A.6 CALCULATION DETAIL OF THE DISTANCE BETWEEN ATOM AND PORE

To quantify the distance of each atom to the nearest pore region, we first identify materials that contain sufficiently large pores. For each crystal structure, we uniformly sample $32 \times 32 \times 32$ points within the lattice using the same grid construction described in Appendix A.2. For every sampled point, we compute its minimum distance to the surrounding atoms. Points whose nearest-

atom distance exceeds the pore threshold $r_{\mathrm{pore}} = 2.0\,\text{Å}$ are designated as *pore points*, representing regions of locally low atomic density.

Once the set of pore points is obtained, we compute for each atom its minimum Euclidean distance to this pore point set. This value is used as the atom's distance-to-pore metric, reflecting how deeply the atom is embedded within dense regions of the structure.

### A.7 DETAILS OF OUT-OF-DISTRIBUTION VALIDATION

To evaluate the out-of-distribution (OOD) performance of SpatialRead, we reorganized the $N_2$ adsorption dataset from CoRE-MOF (Chung et al., 2019). For each structure, we computed its void fraction using `zeo++` (Willems et al., 2012). Materials in the top one-seventh of void fraction were selected as the OOD test set. The remaining six-sevenths were randomly split into training and validation subsets with a $5:1$ ratio, ensuring that the training and validation sets share the same underlying distribution. This setup allows the in-distribution (ID) performance of the model to be assessed independently from its ability to generalize to high-porosity, distribution-shifted MOFs.

Table 9: OOD evaluation on high-porosity MOFs. SpatialRead preserves ranking stability under distribution shift.

| Model | MAE | $R^2$ | Spearman |
|---|---|---|---|
| PaiNN (40 epoch, ID) | 18.7 | 0.907 | 0.961 |
| PaiNN+SpatialRead (3 epoch, ID) | 21.5 | 0.904 | 0.963 |
| PaiNN+SpatialRead (40 epoch, ID) | 11.8 | 0.967 | 0.983 |
| PaiNN (40 epoch, OOD) | 151 | -0.019 | 0.757 |
| PaiNN+SpatialRead (3 epoch, OOD) | 169 | -0.138 | 0.920 |
| PaiNN+SpatialRead (40 epoch, OOD) | 96.0 | 0.526 | 0.951 |

As shown in Table 9, the 3-epoch SpatialRead-enhanced PaiNN model attains comparable validation-set accuracy to the 40-epoch baseline PaiNN model, reflecting the low training cost of our spatial node augmentation. However, both models experience substantial degradation on the OOD test set, as high-porosity structures are absent from the training distribution. Because adsorption capacity is positively correlated with void fraction, the models systematically underestimate adsorption for highly porous materials, leading to large drops in MAE and $R^2$.

Despite this distribution shift, both PaiNN and SpatialRead-enhanced PaiNN preserve the relative ordering of materials, achieving Spearman correlation coefficients of $0.76$ and $0.92$, respectively. This indicates that SpatialRead substantially improves ranking stability under OOD conditions, even when absolute prediction accuracy deteriorates.

### A.8 STATISTICAL ANALYSIS OF THE INTERPRETATION OF SPATIAL NODES

Fig. 5 compares node contributions with the number of atoms contained in their regions. Strikingly, regions with few or no atoms exhibit the largest contributions. This observation is physically consistent: gas molecules cannot be adsorbed into dense atomic regions but are much more likely to be stored in sparse pore regions.

### A.9 EFFECT OF SAMPLE STRATEGY

Table 10: Adaptive sampling of spatial nodes on COFs. Resolution-based sampling uses an average of 462 spatial nodes, compared to 512 in the fixed-grid setting.

| Sampling Strategy | MAE | MSE | $R^2$ |
|---|---|---|---|
| Fixed Grid | 3.52 | 31.0 | 0.979 |
| Resolution-based | 2.31 | 18.6 | 0.987 |

To evaluate whether adaptive sampling improves the representation of spatial regions, we conducted additional experiments on covalent organic frameworks (COFs), which have the largest unit-cell volumes in our dataset and therefore serve as an appropriate test case. Spatial nodes were sampled

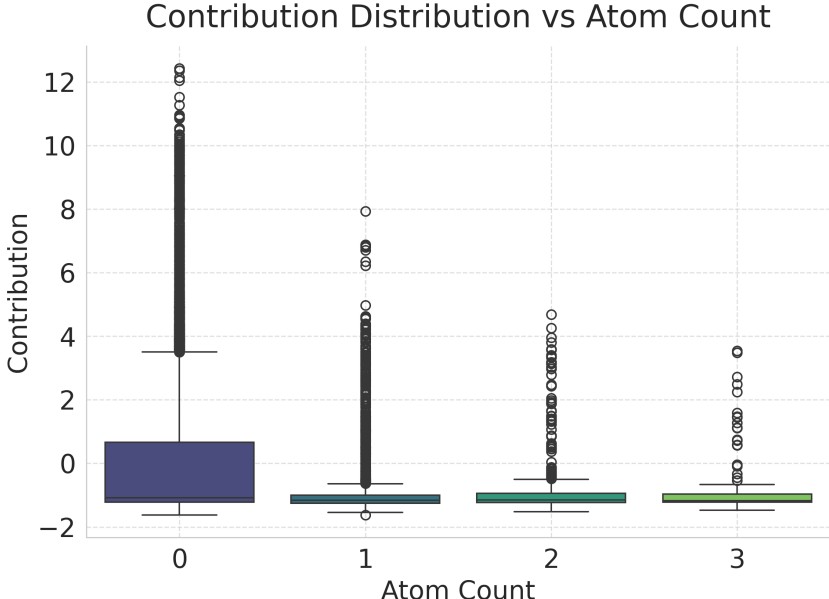

Figure 5: **Atom count (in each region) v.s. contribution.** We counted the number of atoms contained in the area occupied by each spatial node, as well as the contribution of that spatial node.

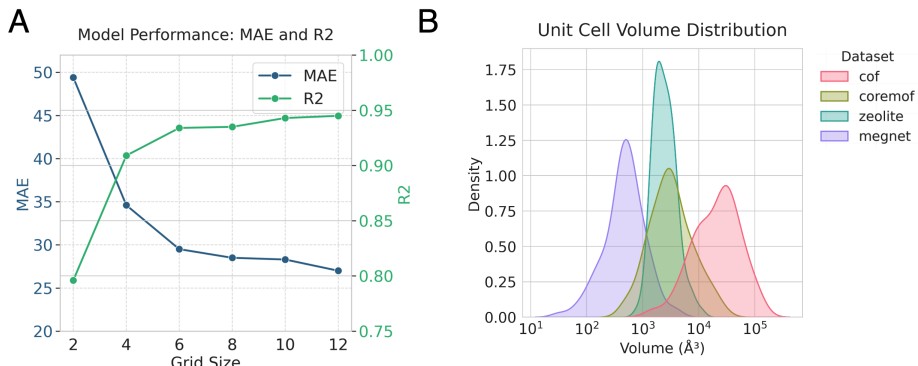

Figure 6: **Effect on sampling ratio. (A) Performance for different number of spatial nodes.** We test number of spatial nodes from $2 * 2 * 2$ to $12 * 12 * 12$. SchNet is adopted to test the best setting of spatial nodes. The corresponding MAE and R2 are drawn in the line plot. **(B) Distribution of volume for different materials.** The volume of a single cell is drawn.

using a resolution-based strategy with a spatial resolution of 4.03 Å, resulting in an average of 462 nodes per structure, compared to the 512 nodes used in the fixed-grid scheme. Despite using fewer nodes, the resolution-based sampling achieves better predictive performance on COFs, indicating that adaptively allocating spatial nodes according to lattice size can improve spatial coverage. We also observe that resolution-based sampling assigns more spatial nodes to larger unit cells, which increases peak memory usage relative to the fixed-grid approach.

### A.10    EFFECT ON SAMPLING RATIO

We considered different numbers of Spatial Node sampling points, ranging from $2*2*2$ to $12*12*12$. The results are presented in Fig. 6. As the number of sampling points increases, the model accuracy gradually improves.

