# OpenReview forum: "From atom to space: A region-based readout function for spatial properties of materials"
_ICLR.cc/2026/Conference — ICLR 2026 Poster_

### Official Review · Reviewer_DEPu · 2025-10-25

**Soundness:** 2
**Presentation:** 2
**Contribution:** 2
**Rating:** 4
**Confidence:** 4

**Summary:**

Introduces a new readout function for materials property prediction tasks using GNNs/MPNNs. The idea of the paper is to introduce a grid of additional spatial nodes and readout functions that only use learned node features of these spatial nodes or readout functions that take into account atom and spatial nodes. Experiments are performed on different datasets, and very good performance is demonstrated across different tasks and datasets, compared to fine-tuned models and models trained from scratch.

**Strengths:**

The experiments show very strong performance of the newly introduced readout functions based on spatial nodes.

The analysis in Figure 2 and the additional experiments in Figure 3 are very interesting and insightful.

**Weaknesses:**

The paper claims that current read-out functions introduce an inductive bias that is not suitable for certain tasks and materials, e.g. the prediction of gas adsorption capacity in metal organic frameworks. This claim is not completely unreasonable, but it is also not supported by any quantifiable evidence. An equivariant GNN can learn local geometric atom environments (also around a pore), up to a cutoff radius which is determined by cutoff-hyperparameters and model depth. Thus, in principle, there is no direct reason to claim that it is impossible to learn pore volumes as a function of mean- or sum-pooled atom embeddings. Every atom can learn from its local geometric environment if it is adjacent to a pore or not, and what pore volume is "attributed" to this atom, so sum pooling can yield the pore volume.

The baseline models CGCNN, GemNet, and PaiNN are more than 5 years old. The performance of newer GNN architectures is not compared. What is the performance of the models that are currently leading the MatBench benchmark?
The pretrained models are newer, but the authors do not compare the performance of the fine-tuned models to the performance if those models were trained from scratch.

Generally, a more differentiated comparison to other, more recent readout functions is missing.

**Questions:**

In Section 2, you discuss readout functions used in the GNN/MPNN literature. All references are older than 2019. Please report about more recent developments of readout functions, e.g. the ones discussed in [1] or even completely different approaches such as [2].
[1] Liu, C., Zhan, Y., Wu, J., Li, C., Du, B., Hu, W., Liu, T. and Tao, D., 2022. Graph pooling for graph neural networks: Progress, challenges, and opportunities. arXiv preprint arXiv:2204.07321.
[2] Wu, Z., Jain, P., Wright, M., Mirhoseini, A., Gonzalez, J.E. and Stoica, I., 2021. Representing long-range context for graph neural networks with global attention. Advances in neural information processing systems, 34, pp.13266-13279.

If Theorem 3.1 is true and Eq. 6 and 8 have the same expressivity, what is the advantage of Eq. 8 over Eq. 6?

What is the difference between extensive quantities of materials and your definition of spatial properties according to definition 3.1? Are all extensive quantities also spatial quantities?

What are the exact definitions of N(s_j) and N(v_j) in Eq. 9 and 10? How many atomic node connections does every spatial node have, and how many spatial node connections? Is this based on cutoff radii as indicated by the phrase "nearby atoms"? How are "adjacent spatial nodes" defined? The 6 nearest neighbor voxels? Or 26? Or also based on a cutoff radius? Figure 1 does not indicate any s_i to s_j edges. How are the positions of the spatial nodes selected? Figure 1 does not indicate a regular lattice. How is the spatial voxel grid defined for more complex space groups than the one shown in Figure 1? Are distances defined in relative coordinates (within the unit cell) or in absolute coordinates (in real space)?

Multi-Modal Transformer: "we impose an explicit ordering" - How is this done, and how does it compare for different types of symmetry groups with different unit cell shapes?

Equation 11: Why does every spatial node only contribute a scalar contribution to the final property? Why do you not use p = MLP_final(Sum(MLP(h))), where MLP(h) is outputting a vector rather than a scalar?

Minor comments: Mistake in Theorem 3.1: "then is must can be".

---

> ### Author Response · Authors · 2025-11-27
> **Response to Reviewer DEPu**
>
> Thanks for taking time to review our paper. We appreciate the concern about the quantifiable evidence of our motivation and the constructive advice on the evaluation methods. We have revised our paper to (1) Quantify the rationality of our motivations. We first demonstrate that, even for original GNN without spatial nodes, the most contributing atoms to the target property are already located near the pores. Next, we evaluate our method via the out-of-distribution (OOD) dataset to demonstrate that our method learns better physically-grounded decomposition and achieves better generalization and more stable prediction. (2) As the reviewer mentioned, we have added a comparison with other readout methods. In addition, the scratch version of the mentioned pre-trained baseline JMP has been added. (3) We revised the paper to make our method more clear, especially the message passing process involving spatial nodes.

---

> > ### Author Response · Authors · 2025-11-27
> >
> > ## Q1
> >
> > The paper claims that current read-out functions introduce an inductive bias that is not suitable for certain tasks and materials, e.g. the prediction of gas adsorption capacity in metal organic frameworks. This claim is not completely unreasonable, but it is also not supported by any quantifiable evidence. An equivariant GNN can learn local geometric atom environments (also around a pore), up to a cutoff radius which is determined by cutoff-hyperparameters and model depth. Thus, in principle, there is no direct reason to claim that it is impossible to learn pore volumes as a function of mean- or sum-pooled atom embeddings. Every atom can learn from its local geometric environment if it is adjacent to a pore or not, and what pore volume is "attributed" to this atom, so sum pooling can yield the pore volume.
> >
> > If Theorem 3.1 is true and Eq. 6 and 8 have the same expressivity, what is the advantage of Eq. 8 over Eq. 6?
> >
> > ## R1
> >
> > We thank the reviewer for the helpful comments. We agree that an equivariant GNN can, in principle, infer whether an atom is located near a pore. To directly examine this, we conducted the following analyses.
> >
> > ### Does a vanilla equivariant GNN automatically focus on pore-adjacent atoms?
> >
> > To validate this point, a PaiNN is trained for gas adsorption prediction. We computed per-atom contributions and compared them to the atom–pore distance. As shown in Table 1, among the top 1% of contributing atoms, nearly 86% are located close to the pore. These results indicate that when predicting spatial properties, the GNN does implicitly learn to identify important regions and distribute the contributions of these regions to the neighboring atoms.
> >
> > Table 1. The proportion of atoms that are close to the pore channels in high-contribution atoms. Detailed calculation method can be found in the revised paper Appendix A.6.
> > | Atom Group                | % Near Pore (<0.05 Å) |
> > | ------------------------- | --------------------- |
> > | Top 1% contributing atoms | 86%               |
> > | All atoms                 | 33%               |
> >
> > We also provide visualization on Fig. 1 in the revised paper, which further confirms that PaiNN indeed places most contribution on atoms near pore surfaces. This supports the reviewer’s point: equivariant GNNs *can* implicitly learn pore-locality without Spatial Nodes.
> >
> > ### Why does the Spatial Node still help?
> >
> > Spatial Node does not increase expressive power. Its benefit is providing an inductive bias aligned with pore-related physics, improving robustness and stability under distribution shift. To validate this, we created an OOD split by placing the top 1/7 highest-porosity MOFs into the test set (see Section 4.9 in the revised paper for detail). In this case, PaiNN (+ Spatial Node) is trained in low-porosity MOFs, validated in low-porosity MOFs (in-distribution) and tested in high-porosity MOFs (out-of-distribution). To ensure a fair comparison, Spatial-PaiNN is trained with fewer epochs such that both models achieve similar in-distribution validation performance.
> >
> > Table 1. Out-of-distribution evaluation for SpatialRead.
> > | Model | MAE | R2 | Spearman |
> > | ----- | --- | -- | -------- |
> > | PaiNN (40 epoch, ID) | 18.7 | 0.907 | 0.961 |
> > | PaiNN + Spatial Node (3 epoch) (ID) | 21.5 | 0.904 | 0.963 |
> > | PaiNN + Spatial Node (40 epoch) (ID) | 11.8 | 0.967 | 0.983 |
> > | PaiNN (40 epoch, OOD) | 151 | -0.0194 | 0.757 |
> > | PaiNN + Spatial Node (3 epoch, OOD) | 169 | -0.138 | 0.920 |
> > | PaiNN + Spatial Node (40 epoch, OOD) | 96.0 | 0.526 | 0.951 |
> >
> > Due to the higher porosity of OOD data compared to ID data, both PaiNN and Spatial-PaiNN exhibit significant underestimation, resulting in deterioration of both MAE and R2 performance. Despite this, both base PaiNN and Spatial-PaiNN maintains the order of the material (with the spearman coefficient larger than 0.7). **Spatial-PaiNN maintains ranking stability much better** (Spearman 0.920 vs. 0.757). Detailed scatter plots are shown in Fig. 4 in the revised paper, which also show that vanilla PaiNN exhibits significantly larger prediction variance, leading to unstable ordering.
> >
> > While PaiNN already learns to focus on pore-adjacent atoms, **explicitly encoding this structure via Spatial Nodes produces a more stable and robust inductive bias**, especially under distribution shift. This is precisely where Spatial Nodes provide practical value.

---

> > > ### Author Response · Authors · 2025-11-27
> > >
> > > ## Q2
> > >
> > > The baseline models CGCNN, GemNet, and PaiNN are more than 5 years old. The performance of newer GNN architectures is not compared. What is the performance of the models that are currently leading the MatBench benchmark? The pretrained models are newer, but the authors do not compare the performance of the fine-tuned models to the performance if those models were trained from scratch.
> > >
> > > ## R2
> > >
> > > Thanks for the constructive suggestion. As the reviewer and other reviewers mentioned, the benchmark setting of our paper may reduces the credibility of the work. First, we would like to point out that JMP [1] is already the leading neural network on the MatBench benchmark. JMP is a pre-trained GemNet[2] on large-scale molecular/material datasets. Therefore, its scratch version (GemNet) is already provided in Table 1 of the original paper. JMP outperforms GemNet in the vast majority of tasks, demonstrating significant effects in large-scale pre-training.
> > >
> > > [1] Shoghi, N., Kolluru, A., Kitchin, J. R., Ulissi, Z. W., Zitnick, C. L., & Wood, B. M. From Molecules to Materials: Pre-training Large Generalizable Models for Atomic Property Prediction. In The Twelfth International Conference on Learning Representations.
> > >
> > > [2] Gasteiger, J., Becker, F., & Günnemann, S. (2021). Gemnet: Universal directional graph neural networks for molecules. Advances in Neural Information Processing Systems, 34, 6790-6802.
> > >
> > > ## Q3
> > >
> > > Generally, a more differentiated comparison to other, more recent readout functions is missing.
> > >
> > > In Section 2, you discuss readout functions used in the GNN/MPNN literature. All references are older than 2019. Please report about more recent developments of readout functions, e.g. the ones discussed in [1] or even completely different approaches such as [2]. [1] Liu, C., Zhan, Y., Wu, J., Li, C., Du, B., Hu, W., Liu, T. and Tao, D., 2022. Graph pooling for graph neural networks: Progress, challenges, and opportunities. arXiv preprint arXiv:2204.07321. [2] Wu, Z., Jain, P., Wright, M., Mirhoseini, A., Gonzalez, J.E. and Stoica, I., 2021. Representing long-range context for graph neural networks with global attention. Advances in neural information processing systems, 34, pp.13266-13279.
> > >
> > > ## R3
> > >
> > > Thank you for the suggestion regarding recent developments in readout functions. We address this point from two perspectives. In addition, we have added a short discussion of recent pooling and readout techniques in the Related Works section.
> > >
> > > ### 1. Rationale for using simple additive pooling in molecular and materials GNNs
> > >
> > > First, we want to point out the reason we do not compare our method with other readout functions. In the field of property prediction, most GNNs adopt sum or mean pooling as the readout. Examples include SchNet (NIPS 2017), DimeNet (NIPS 2020), PaiNN (ICML 2021), GemNet (NIPS 2021), ViSNet (2024, Nature Communications), and recent deep-learning potentials such as MACE (NIPS 2022). This choice is standard for two reasons: (i) many target properties in this domain, such as total energies, are atom-decomposable and therefore naturally compatible with sum/mean pooling; and (ii) molecular and materials graphs are generally small and lack the hierarchical or community structures for which clustering-based pooling methods are typically designed. As a result, although hierarchical pooling has been extensively explored in generic graph learning settings, it is rarely adopted in molecular and materials prediction.
> > >
> > > ### 2. Additional evaluation of more recent readout mechanisms
> > >
> > > To directly address the reviewer’s request, we conducted experiments using representative modern readout modules that are applicable to scalar prediction, including GraphTrans[1] and GMT[2]. Both methods provide moderate improvements over the baseline. Our SpatialRead module, however, consistently yields the largest gains across all datasets. Results are shown below.
> > >
> > > Table 1. Comparison to other readout functions.
> > > | Model                   | MOF C3H6/C3H8 | MOF N2 | MOF CH4/N2 | COF CH4 | PPN CH4 | Zeolite heat |
> > > | ----------------------- | ------------- | ------ | ---------- | ------- | ------- | ------------ |
> > > | PaiNN                   | 0.691         | 0.979  | 0.867      | 0.736   | 0.856   | 0.791        |
> > > | PaiNN + GraphTrans | 0.712         | 0.912  | 0.870      | 0.750   | 0.847   | 0.801        |
> > > | PaiNN + GMT             | 0.740         | 0.924  | 0.866      | 0.742   | 0.863   | 0.803        |
> > > | PaiNN + SpatialRead     | 0.784         | 0.987  | 0.941      | 0.987   | 0.977   | 0.969        |
> > >
> > > [1] Wu, Z., Jain, P., Wright, M., Mirhoseini, A., Gonzalez, J. E., & Stoica, I. (2021). Representing long-range context for graph neural networks with global attention. Advances in neural information processing systems, 34, 13266-13279.
> > >
> > > [2] Baek, J., Kang, M., & Hwang, S. J. (2021). ACCURATE LEARNING OF GRAPH REPRESENTATIONS WITH GRAPH MULTISET POOLING. In 9th International Conference on Learning Representations, ICLR 2021.

---

> > > > ### Author Response · Authors · 2025-11-27
> > > >
> > > > ## Q4
> > > >
> > > > What is the difference between extensive quantities of materials and your definition of spatial properties according to definition 3.1? Are all extensive quantities also spatial quantities?
> > > >
> > > > ## R4
> > > >
> > > > Spatial properties in our framework are defined solely by whether a quantity can be meaningfully decomposed over explicit spatial regions (Definition 3.1). This notion is independent of whether the quantity is extensive or intensive. We clarify this with four examples:
> > > >
> > > > - Adsorption capacity (cm³/g, note the unit, which is the unit used in our experiment, Table 1) — intensive and spatial: although normalized per gram, its value results from contributions of different pore regions.
> > > > - Pore volume (cm³) — extensive and spatial: it can be decomposed over different pore channels or cavities.
> > > > - Total mass (g) — extensive but not spatial: while it scales with system size, it does not admit a meaningful attribution to specific spatial regions in the sense of Definition 3.1.
> > > > - Bandgap (eV) — intensive and not spatial: it is a global electronic property and cannot be decomposed over spatial cells to recover the global value.
> > > >
> > > > Despite this, we acknowledge that providing a perfectly sharp definition of spatial properties is difficult. As discussed in the paper, the region-decomposable and atom-decomposable representations is mathematically equivalent. The key distinction, therefore, is not whether a property can be decomposed over spatial regions in principle, but whether such a spatial decomposition is natural and appropriate for the physical fact.
> > > >
> > > > ## Q5
> > > >
> > > > What are the exact definitions of N(s_j) and N(v_j) in Eq. 9 and 10? How many atomic node connections does every spatial node have, and how many spatial node connections? Is this based on cutoff radii as indicated by the phrase "nearby atoms"? How are "adjacent spatial nodes" defined? The 6 nearest neighbor voxels? Or 26? Or also based on a cutoff radius? Figure 1 does not indicate any s_i to s_j edges. How are the positions of the spatial nodes selected? Figure 1 does not indicate a regular lattice. How is the spatial voxel grid defined for more complex space groups than the one shown in Figure 1? Are distances defined in relative coordinates (within the unit cell) or in absolute coordinates (in real space)?
> > > >
> > > > ## R5
> > > >
> > > > We would like to express our gratitude to the reviewers for taking the time to review our paper. We are grateful for their detailed suggestions. We have realized that our method description was not detailed enough. We have revised the paper to clearly explain the specific details of our spatial nodes. The following is the detailed response to the reviewers' comments.
> > > >
> > > > (1) Definitions of $N(s_j)$ and $N(v_j)$.
> > > >
> > > > In our actual implementation: $N(s_j)$ denotes the set of atomic neighbors of spatial node $s_j$ (i.e., atoms connected to $s_j$ by an edge). $N(v_j)$ denotes the set of atomic neighbors of atom $v_j$, identical to standard atom–atom neighborhoods. There are no edges between spatial nodes in the instantiated model used in our experiments. The interaction between spatial nodes is achieved by the later Multi Modal Transformer module.
> > > >
> > > > (2) Why there is no message passing between spatial nodes.
> > > >
> > > > The spatial–spatial message passing is an optional component. Some spatial properties may require spatial–spatial interactions. In this work, we disable this term, i.e. $N_s(s_j)=\varnothing$, because:
> > > >
> > > > - gas adsorption does not require explicit local coupling between spatial regions, and
> > > > - the global readout module already integrates information across spatial nodes.
> > > >
> > > > (3) Neighbor construction and cutoff radii.
> > > >
> > > > Atom–atom and atom–spatial edges are built using the same radius-based neighborhood:
> > > >
> > > > - search all atoms within cutoff $r_{\text{cut}}$,
> > > > - retain the nearest ones up to max\_num\_neighbors.
> > > >   Thus, each spatial node may connect to a variable number of atoms, and only atom–atom and atom–spatial edges exist.
> > > >
> > > > (4) Placement of spatial nodes.
> > > >
> > > > Spatial nodes are generated by grid sampling in fractional coordinates of the unit cell:
> > > >
> > > > $$
> > > > \left(\frac{i}{G},\frac{j}{G},\frac{k}{G}\right), \quad i,j,k=0,\dots,G-1,\ \text{with}\ G^3 = num \_ spnode.
> > > > $$
> > > >
> > > > These points are then mapped to Cartesian coordinates via the lattice vectors.
> > > > This procedure naturally supports arbitrary lattice geometries and space groups; it does not assume a cubic or regular lattice.
> > > >
> > > > (5) Coordinate system for distance computation.
> > > >
> > > > All distances (atom–atom and atom–spatial) are computed in Cartesian coordinates, following the standard practice in materials/moecules GNNs, with periodic boundary conditions handled by the neighbor search.
> > > >
> > > > We have integrated these clarifications into the revised paper, mainly in Appendix A.2 due to space limitation.

---

> > > > > ### Author Response · Authors · 2025-11-27
> > > > >
> > > > > ## Q6
> > > > >
> > > > > Multi-Modal Transformer: "we impose an explicit ordering" - How is this done, and how does it compare for different types of symmetry groups with different unit cell shapes?
> > > > >
> > > > > ## R6
> > > > >
> > > > > We thank the reviewer for raising this point. The “explicit ordering” refers to the fact that spatial nodes are generated by a deterministic 3D grid sampling in fractional coordinates of the unit cell, indexed by a fixed triplet $(i,j,k)$ with $i,j,k \in {0,\dots,G-1}$. This ordering is therefore intrinsic to the sampling procedure and does not depend on the shape of the unit cell. After sampling, each node is mapped to Cartesian coordinates through the lattice vectors, but its grid index remains unchanged, which provides a canonical ordering for the Multi-Modal Transformer.
> > > > >
> > > > > For the Transformer, we use a simple learnable positional embedding: an embedding table of size $G^3 \times F$ (where $G^3$ is the number of spatial nodes) added to the spatial node features. Because the grid indexing is defined in fractional space, this positional embedding works uniformly across crystals with different lattice geometries and space groups. Our focus is not on encoding crystallographic symmetries; therefore, we adopt this straightforward positional encoding.
> > > > >
> > > > > ## Q7
> > > > >
> > > > > Equation 11: Why does every spatial node only contribute a scalar contribution to the final property? Why do you not use p = MLP_final(Sum(MLP(h))), where MLP(h) is outputting a vector rather than a scalar?
> > > > >
> > > > > ## R7
> > > > >
> > > > > Thank you for the question. This design actually follows JMP[1], where each node produces a scalar contribution and the global property is obtained through summation. This choice is motivated by JMP's strong empirical generalization across many material and molecular benchmarks. In addition, in the reply to Reviewer 4Vi5, we have provided the effect of applying both Jumping Knowledge Connection[2] and the mentioned $p = MLP_final(Sum(MLP(h)))$. In summary, the improvement is marginal.
> > > > >
> > > > > Table 1. Performance of applying cross-layer information aggregation and the mentioned readout on PaiNN with Spatial Nodes
> > > > > | Model | MOF C3H6/C3H8 sep. | MOF N2 ads. | MOF CH4/N2 sep. | COF CH4 ads. | PPN CH4 ads. | zeolite CH4 heat. |
> > > > > | ----- | -------------- | ------- | ----------- | -------- | -------- | ----------------- |
> > > > > | PaiNN+SpNode | 0.794 | 0.978 | 0.936 | 0.979 | 0.978 | 0.886 |
> > > > > | PaiNN+SpNode+JumpingKnowledge | 0.806 | 0.979 | 0.944 | 0.979 | 0.979 | 0.890 |
> > > > >
> > > > > [1] Shoghi, N., Kolluru, A., Kitchin, J. R., Ulissi, Z. W., Zitnick, C. L., & Wood, B. M. From Molecules to Materials: Pre-training Large Generalizable Models for Atomic Property Prediction. In The Twelfth International Conference on Learning Representations.
> > > > >
> > > > > [2] Xu, K., Li, C., Tian, Y., Sonobe, T., Kawarabayashi, K. I., & Jegelka, S. (2018, July). Representation learning on graphs with jumping knowledge networks. In International conference on machine learning (pp. 5453-5462). pmlr.
> > > > >
> > > > > ## Q8
> > > > >
> > > > > Minor comments: Mistake in Theorem 3.1: "then is must can be".
> > > > >
> > > > > ## R8
> > > > >
> > > > > Thank you for the constructive suggestion. We have revised our paper accordingly.

---

### Official Review · Reviewer_G88g · 2025-10-26

**Soundness:** 3
**Presentation:** 3
**Contribution:** 3
**Rating:** 8
**Confidence:** 5

**Summary:**

This paper identifies a key inductive bias limitation in graph neural networks (GNNs) for material property prediction — namely, that existing message passing–readout frameworks assume atom-decomposable properties, which is not valid for spatially defined material properties such as adsorption, separation, or accessible surface area.

To overcome this, the authors propose SpatialRead, a novel readout function that reformulates the graph-level representation as an integral over space rather than a sum over atoms. The method introduces spatial nodes that voxelize the material domain, forming a heterogeneous atom–space graph with one-way message flow from atoms to spatial nodes. A Transformer-based multimodal readout fuses atomic and spatial representations.

Theoretically, the authors prove that the space-integral and node-summation formulations are equivalent in expressivity when the receptive field is finite. Empirically, SpatialRead achieves state-of-the-art results on 44,157 porous materials and 27 spatial-property prediction tasks, outperforming even large-scale pre-trained foundation models such as JMP (120M samples). The approach also maintains strong performance on conventional (non-spatial) material properties.

**Strengths:**

- The first formal unification of node-decomposable and region-decomposable graph readouts.
- The directionality (atoms → space) aligns with actual field theory and adsorption physics.
- Equivalence theorem ensures expressivity preservation.
- 44k+ samples and 27 tasks, across multiple material systems.
- ~2.9 MB model surpasses 38 MB foundation models.
- Bridges discrete atomic graphs and continuous spatial modeling paradigms.

**Weaknesses:**

- The voxelization scheme (e.g., 8×8×8) is fixed; adaptive resolution or hierarchical partitioning could further improve scalability and accuracy?
- Although SpatialRead outperforms pretrained models from scratch, it would be interesting to see whether adding pretraining further enhances its capability.
-  The paper focuses on porous materials and spatial properties; it remains unclear whether the proposed readout function generalizes to conventional datasets like Materials Project or JARVIS, especially since the baseline models (e.g., CGCNN, ALIGNN, Matformer) were originally designed and benchmarked on such conventional crystal datasets.

**Questions:**

- Does the voxelization scheme preserve lattice periodicity? If not, predictions might vary under cell replication.
- While efficient for moderate datasets, its scalability to extremely large systems (>10⁴ atoms) isn’t thoroughly discussed.

---

> ### Author Response · Authors · 2025-11-27
> **Response to Reviewer G88g**
>
> ## Q1
>
> The voxelization scheme (e.g., 8×8×8) is fixed; adaptive resolution or hierarchical partitioning could further improve scalability and accuracy?
>
> ## R1
>
> Thanks for the constructive suggestion. To examine the effect of adaptive sampling, we conducted additional experiments on COFs, which have the largest unit-cell volumes in our dataset and therefore provide a suitable test case for evaluating whether resolution-based spatial sampling is beneficial. The results are shown below. Although the resolution-based method uses fewer spatial nodes on average (462 vs. 512 in the fixed-grid setting), it achieves better performance on COFs. Nevertheless, we also note that resolution-based sampling assigns more spatial nodes to larger structures, which leads to higher peak memory usage compared to fixed-grid sampling.
>
> Table 1. Adaptive sampling of spatial nodes on COFs. Spatial nodes are sampled at a spatial resolution of 4.03 Å, resulting in 462 nodes on average, fewer than the 512 nodes used in the fixed-grid sampling.
>
> | Sampling | MAE | MSE | R2 |
> | -------- | --- | --- | -- |
> | Fixed Grid | 3.52 | 31.0 | 0.979 |
> | Resolution | 2.31 | 18.6 | 0.987 |
>
> ## Q2
>
> Although SpatialRead outperforms pretrained models from scratch, it would be interesting to see whether adding pretraining further enhances its capability.
>
> ## R2
>
> Thanks for the constructive suggestions. We have applied our method (Spatial Node + Multi Modal Transformer) directly to the already-pre-trained JMP[1]. Surprisingly, although JMP is pre-trained without spatial nodes, our method produces consistent improvement. This demonstrates the universality of our method as a plug-and-play module. In addition, we also observed that our method benefits from pre-training. The performance of the scratch version of JMP (i.e. GemNet) significantly lags behind JMP, either with or without SpatialRead.
>
> However, we have observed that JMP + SpatialRead did not yield better results than PaiNN + SpatialRead. We emphasize that this is probably a limitation of the computational load. Due to the consideration of three-body and four-body interactions as well as higher embedding and edge feature encoding, the size of the memory usage and the training load increase rapidly with the number of neighbors. For actual MOFs like in the CoREMOF[2] dataset, the NVIDIA RTX GeForce 4090 (24 GB) only allows us to set the maximum value of max_num_neighbors to 15 (as a comparison, the original setting of JMP in MOFs set the max_num_neighbors to 5). Properties such as adsorption capacity are significantly influenced by the intermolecular interactions, and therefore may be more sensitive to parameters like cutoff and max_num_neighbors. Even under such a disadvantage, JMP still achieved performance comparable to that of PaiNN. It can be expected that if the complete 30 maximum neighbors are enabled, the effect of JMP will surpass that of PaiNN. However, due to the computational burden and time constraints, we did not conduct this experiment. As for PaiNN, due to its simple two-body message passing process, we can allow each atom to have up to 30 neighboring nodes.
>
> Table 1. Performance (R2 score) of applying SpatialRead on the already-pre-trained model JMP.
> | Model | MOF C3H6/C3H8 sep. | MOF N2 ads. | MOF CH4/N2 sep. | COF CH4 ads. | PPN CH4 ads. | zeolite CH4 heat. |
> | ----- | -------------- | ------- | ----------- | -------- | -------- | ----------------- |
> | PaiNN+SpatialRead | 0.784 | 0.987 | 0.941 | 0.987 | 0.977 | 0.969 |
> | GemNet | 0.729 | 0.968 | 0.924 | 0.816 | 0.932 | 0.836 |
> | GemNet+SpatialRead | 0.753 | 0.979 | 0.921 | 0.986 | 0.923 | 0.881 |
> | JMP | 0.774 | 0.971 | 0.908 | 0.884 | 0.947 | 0.874 |
> | JMP+SpatialRead | 0.792 | 0.988 | 0.941 | 0.982 | 0.969 | 0.945 |
>
> [1] Shoghi, N., Kolluru, A., Kitchin, J. R., Ulissi, Z. W., Zitnick, C. L., & Wood, B. M. From Molecules to Materials: Pre-training Large Generalizable Models for Atomic Property Prediction. In The Twelfth International Conference on Learning Representations.
>
> [2] Chung, Y. G., Haldoupis, E., Bucior, B. J., Haranczyk, M., Lee, S., Zhang, H., ... & Snurr, R. Q. (2019). Advances, updates, and analytics for the computation-ready, experimental metal–organic framework database: CoRE MOF 2019. Journal of Chemical & Engineering Data, 64(12), 5985-5998.

---

> > ### Author Response · Authors · 2025-11-27
> >
> > ## Q3
> >
> > The paper focuses on porous materials and spatial properties; it remains unclear whether the proposed readout function generalizes to conventional datasets like Materials Project or JARVIS, especially since the baseline models (e.g., CGCNN, ALIGNN, Matformer) were originally designed and benchmarked on such conventional crystal datasets.
> >
> > ## R3
> >
> > Table 1. Full results in the MatBench[1] dataset. MAE is adopted as the metric, following JMP.
> > | Model | JDFT2D | Phonons | Dielectric | Log GVRH | Log KVRH | Perovskites | MP Gap | MP E Form |
> > | ----- | ------ | ------- | ---------- | -------- | -------- | ----------- | ------ | --------- |
> > | MODNet | 25.55 | 34.77 | 0.169 | 0.073 | 0.054 | 0.093 | 0.215 | 40.2 |
> > | coGN | 22.25 | 32.12 | 0.178 | 0.068 | 0.052 | 0.027 | 0.153 | 17.4 |
> > | JMP | 20.72 | 26.6 | 0.133 | 0.06 | 0.044 | 0.029 | 0.119 | 13.6 |
> > | JMP + SpatialRead | 18.17 | 25.8 | 0.133 | 0.06 | 0.047 | 0.030 | 0.107 | 15.3 |
> >
> > Thank you for the suggestion. We have added evaluations on the full MatBench benchmark. When SpatialRead is applied to a strong backbone JMP[2], the performance is comparable to the original JMP. For tasks that clearly follow an extensive, atom-wise additive form, such as formation energy, sum pooling already provides the most appropriate inductive bias, and in these cases JMP combined with SpatialRead performs slightly worse. For tasks that do not obviously satisfy atom-wise additivity, SpatialRead offers a more flexible readout and maintains comparable performance. In addition, band gap depends on uneven contributions from different atoms (the energy difference between the valence band maximum and the conduction band minimum), thus may be not suited for sum-pooling. As a result, the use of attention mechanism in SpatialRead achieves better performance.
> >
> > In conclusion, these results highlight the importance of selecting a readout aligned with the underlying physical structure of the target property. For tasks that clearly conform to either sum pooling or spatial pooling, the corresponding readout should be used to provide an appropriate inductive bias. For properties that do not admit an obvious or physically meaningful decomposition, a Transformer-based readout offers a flexible and generally reliable alternative by leveraging the expressive power of attention.
> >
> > [1] Dunn, A., Wang, Q., Ganose, A., Dopp, D., & Jain, A. (2020). Benchmarking materials property prediction methods: the Matbench test set and Automatminer reference algorithm. npj Computational Materials, 6(1), 138.
> >
> > [2] Shoghi, N., Kolluru, A., Kitchin, J. R., Ulissi, Z. W., Zitnick, C. L., & Wood, B. M. From Molecules to Materials: Pre-training Large Generalizable Models for Atomic Property Prediction. In The Twelfth International Conference on Learning Representations.
> >
> > ## Q4
> >
> > Does the voxelization scheme preserve lattice periodicity? If not, predictions might vary under cell replication.
> >
> > ## R4
> >
> > Spatial nodes are generated by **uniform grid sampling in fractional coordinates** of the unit cell:
> >
> > $$
> > \left(\frac{i}{G},\frac{j}{G},\frac{k}{G}\right), \quad i,j,k=0,\dots,G-1,\ \text{with}\ G^3 = num\_spnode.
> > $$
> >
> > When the number of spatial nodes is fixed, the periodicity will be disrupted. We attempted to construct supercells (2\*1\*1, 1\*2\*1, and 1\*1\*2) for each material, and then evaluated the performance of PaiNN+SpatialRead trained in the original cell. The model performance is indeed compromised. These results emphasize the necessity of designing more advanced sampling methods that are in line with the periodicity. It can be observed that when the lattice expands along the a and b directions, the performance degradation is relatively small. However, when the lattice expands along the c direction, the performance degradation becomes more severe.
> >
> > Table 1. Performance on supercell for the MOF N2 adsorption task.
> > | Performance | 1\*1\*1 | 2\*1\*1 | 1\*2\*1 | 1\*1\*2 |
> > | ------- | ------- | ------- | ------- | ------- |
> > | R2 | 0.987  | 0.973 | 0.970 | 0.951 |

---

> > > ### Author Response · Authors · 2025-11-27
> > >
> > > ## Q5
> > >
> > > While efficient for moderate datasets, its scalability to extremely large systems (>10⁴ atoms) isn’t thoroughly discussed.
> > >
> > > ## R5
> > >
> > > Thanks for the constructive suggestion. Firstly, we emphasize that the complexity brought by a spatial node is similar to that of an additional atom. In our practice, the neighboring nodes are also constructed based on cutoff, consistent with the atoms. Due to the limitation of cutoff, the complexity brought by $M$ spatial nodes does not increase as the system expands. For example, for a model like PaiNN with a complexity of $O(nk)$ (where $n$ is the number of atoms and $k$ is the number of neighbors), the additional complexity brought by $M$ spatial nodes is still $O(Mk)$, independent of $n$. However, the complexity of the used Attention-based heads is $O(n^2+M^2)$. Since the Transformer model for long sequences is not the main focus of this paper, we only report the changes in memory usage of PaiNN + Spatial Node method here.
> > >
> > > As shown, the computational cost of spatial nodes remain as a const. When the number of spatial nodes is fixed, as the system size increases, the space complexity and time complexity brought by the spatial nodes become relatively less significant. However, we also emphasize that for larger systems, more spatial nodes will be needed to ensure sufficient resolution. Therefore, the number of spatial nodes should increase proportionally as the system size increases.
> > >
> > > Table 1. Space and Time complexity for larger system size.
> > > | System Size (number of atoms) | Model | Training Time per epoch / min | Memory / MB |
> > > | ----------- | --- | ------------------------- | ------ |
> > > | 294 | PaiNN | 4.07 | 484 |
> > > | 294 | PaiNN+SpNode | 5.21 | 546=484+62 |
> > > | 634 | PaiNN | 4.86 | 752 |
> > > | 634 | PaiNN+SpNode | 5.83 | 819=752+67 |
> > > | 3092 | PaiNN | 10.7 | 2475 |
> > > | 3092 | PaiNN+SpNode | 11.3 | 2520=2475+45 |
> > > | 8476 | PaiNN | 29.2 | 5857 |
> > > | 8476 | PaiNN+SpNode | 26.4 | 5896=5857+39 |

---

### Official Review · Reviewer_LnWH · 2025-10-31

**Soundness:** 2
**Presentation:** 3
**Contribution:** 2
**Rating:** 2
**Confidence:** 4

**Summary:**

This paper proposes a novel spatial readout mechanism for Graph Neural Networks, with applications to materials science tasks, particularly Metal-Organic Frameworks (MOFs). The authors argue that pooling/readout mechanisms in GNNs still have room for innovation and introduce a method that incorporates additional “spatial” nodes (mostly in void/vacuum) space to improve property predictions.

**Strengths:**

- Addresses a relevant problem: readout mechanisms in GNNs remain an area where improvements are possible
- Applies methodology to practical materials science applications

**Weaknesses:**

### 1. Weak Empirical Validation

The experimental evaluation is insufficient to support the claims:

- **Trivial target properties**: The main benchmarks focus on predicting surface area and pore volume for MOFs. From a materials science perspective, these properties are almost directly derivable from the structure itself (via van der Waals sphere calculations), making them poor choices for demonstrating the value of a novel readout mechanism.

- **Selective MatBench reporting**: The paper only reports results on a subset of MatBench properties. This selective reporting raises concerns about the method's general applicability. What happened to the other MatBench tasks?

- **Inconsistent and marginal improvements**: Across benchmarks, the improvements are neither consistent nor substantial enough to justify the added complexity.

### 2. Missing Ablation Studies

The paper lacks essential ablation studies to disentangle the contributions of different components:

- **Effect of additional void nodes**: The method adds nodes in empty/void space, which allows the model to trivially compute the ratio between occupied and unoccupied space—essentially providing a direct signal for pore accessible volume fraction. This could be the primary driver of any improvements, independent of the readout mechanism.

- **Required ablation**: Compare additional void nodes with standard pooling (max/mean) versus the proposed spatial readout. This would demonstrate whether the gains come from the readout innovation or simply from the additional structural information.

### 3. Added Complexity Without Clear Justification

- The method introduces an additional hyperparameter (sampling degree for void space), increasing model complexity and optimization difficulty
- Given the weak empirical gains, it's unclear whether this added complexity is warranted

### 4. Questionable Benchmark Choice

The authors introduce a new benchmark, but the field already has established benchmarks (e.g., mofdscribe for MOFs). The rationale for a new benchmark is not clearly articulated.

**Questions:**

1. Can you provide complete MatBench results for all properties rather than a selective subset?
2. Can you include an ablation study comparing: (a) base model, (b) base model + void nodes + simple pooling, (c) base model + void nodes + proposed readout?
3. Why is a new benchmark needed when established ones like mofdscribe exist?

---

> ### Author Response · Authors · 2025-11-27
> **Response to Reviewer LnWH**
>
> We thank the reviewer for taking the time to read our paper. The reviewer raises a few concerns about the benchmark setting, ablation study, and hyperparameter optimization. We have taken these constructive suggestions into consideration and revised our paper. To avoid the possible misunderstanding due to our previous writing, before we demonstrate detailed revised experiments about benchmark and ablation, we want to first clarify the main innovation of our work.
>
> SpatialRead is designed mainly to **enhance** current GNNs in the so-called spatial properties (adsorption capacity and separation ratio, etc.) while **maintaining** the performance of GNNs for other tasks (such as those in the MatBench[1] dataset), thus broaden the application scope of GNNs. Such spatial properties, can be easily separated into contributions of each spatial regions. Based on this view, SpatialRead mainly consists of two components:
>
> (1) Auxiliary spatial nodes that provide region-level representations, thus enhancing GNNs on spatial properties
> (2) An attention-based multimodal readout that adaptively combines atomic and spatial features, thus maintaining comparable performance of the base GNN on other non-spatial properties.
>
> In this view, spatial nodes are part of the readout design, analogous to how additional nodes are often considered part of the readout/aggregation stage in prior work[2]. We emphasize the difference between our spatial nodes with previously proposed methods such as virtual node or master node.
>
> |        | Virtual Node  | Spatial Node (Sp Node)                |
> | -------- | ----------------------- | ------------------------------- |
> | Meaning       | Commonly represent a **long-range** context to read and write, visible to all/clustered atoms [2, 3, 4]         | the contribution of the **local** region to the target property (Fig. 2 B)      |
> | Have physical coordinate       | No          | Yes      |
> | Message Passing direction     | Atom <-> VN         | Atom -> Sp Node (unidirectional)                   |
> | Whether change the feature of atoms | Yes            | No, adding spatial nodes do not change the feature of atoms due to undirectional message passing                        |
> | Number of nodes       | Commonly fixed to 1[4]. Large number of VNs can result in performance loss [3]                 | Up to 1728 evaluated in this work                      |
>
> [1] Dunn, A., Wang, Q., Ganose, A., Dopp, D., & Jain, A. (2020). Benchmarking materials property prediction methods: the Matbench test set and Automatminer reference algorithm. npj Computational Materials, 6(1), 138.
>
> [2] Wu, Z., Jain, P., Wright, M., Mirhoseini, A., Gonzalez, J. E., & Stoica, I. (2021). Representing long-range context for graph neural networks with global attention. Advances in neural information processing systems, 34, 13266-13279.
>
> [3] Hwang, E., Thost, V., Dasgupta, S. S., & Ma, T. (2022). An analysis of virtual nodes in graph neural networks for link prediction. In The first learning on graphs conference.
>
> [4] Gilmer, J., Schoenholz, S. S., Riley, P. F., Vinyals, O., & Dahl, G. E. (2017, July). Neural message passing for quantum chemistry. In International conference on machine learning (pp. 1263-1272). Pmlr.

---

> > ### Author Response · Authors · 2025-11-27
> >
> > ## Q1
> >
> > Weak Empirical Validation: Trivial target properties: The main benchmarks focus on predicting surface area and pore volume for MOFs. From a materials science perspective, these properties are almost directly derivable from the structure itself (via van der Waals sphere calculations), making them poor choices for demonstrating the value of a novel readout mechanism.
> >
> > ## R1
> >
> > We appreciate the reviewer’s concern regarding the relevance of the evaluated properties. We would like to clarify that our work does not focus on predicting trivial geometric quantities such as surface area or pore volume. Instead, the primary motivation of our method focus on properties like gas adsorption capacity and separation selectivity, which represent key application scenarios for porous materials (e.g., CO₂ capture, hydrogen storage, methane purification). Experiments reported in Table 1 focus on six representative, non-trivial properties including adsorption capacity, adsorption heat, separation ratio, etc., which are all critical properties for porous materials such as MOFs [1,2], and are mostly obtained from experiments or expensive simulations.
> >
> > In addition, in our benchmark construction, we intentionally separated simple geometric property prediction tasks from more practically meaningful ones. As illustrated in Appendix Figure 5 (in the revised paper, i.e. Appendix Fig. 4 in original paper), the benchmark contains five basic geometric tasks and about twenty functional property prediction tasks. We hope this clarification helps to address the reviewer’s concern regarding the empirical validation.
> >
> > [1] Boyd, P. G., Chidambaram, A., García-Díez, E., Ireland, C. P., Daff, T. D., Bounds, R., ... & Smit, B. (2019). Data-driven design of metal–organic frameworks for wet flue gas CO2 capture. Nature, 576(7786), 253-256.
> > [2] Kim, E. J., Siegelman, R. L., Jiang, H. Z., Forse, A. C., Lee, J. H., Martell, J. D., ... & Long, J. R. (2020). Cooperative carbon capture and steam regeneration with tetraamine-appended metal–organic frameworks. Science, 369(6502), 392-396.
> >
> > ## Q2
> >
> > Selective MatBench reporting: The paper only reports results on a subset of MatBench properties. This selective reporting raises concerns about the method's general applicability. What happened to the other MatBench tasks?
> >
> > ## R2
> >
> > Table 1. Performance of applying SpatialRead on JMP on the full MatBench[1] dataset, compared to the leading network in the leaderboard on the official website.
> > | Model | JDFT2D | Phonons | Dielectric | Log GVRH | Log KVRH | Perovskites | MP Gap | MP E Form |
> > | ----- | ------ | ------- | ---------- | -------- | -------- | ----------- | ------ | --------- |
> > | MODNet | 25.55 | 34.77 | 0.169 | 0.073 | 0.054 | 0.093 | 0.215 | 40.2 |
> > | coGN | 22.25 | 32.12 | 0.178 | 0.068 | 0.052 | 0.027 | 0.153 | 17.4 |
> > | JMP | 20.72 | 26.6 | 0.133 | 0.06 | 0.044 | 0.029 | 0.119 | 13.6 |
> > | JMP + SpatialRead | 18.17 | 25.8 | 0.133 | 0.06 | 0.047 | 0.030 | 0.107 | 15.3 |
> >
> > Thank you for the constructive advice. We have now included the complete MatBench results in Table 1 following JMP[2] for a more comprehensive evaluation. As stated in the paper, the primary goal of SpatialRead is to enhance GNN performance on spatial properties while preserving performance on non-spatial ones. We therefore use the state-of-the-art method JMP[2] on MatBench as the baseline and apply SpatialRead to assess whether it introduces any performance degradation.
> >
> > As shown in Table 1, SpatialRead can be directly applied to state-of-the-art pretrained models without additional re-training, improving spatial-attribute prediction while maintaining performance on general material properties. Overall, SpatialRead has minimal impact on JMP. For properties that can be expressed as a sum of atomic contributions (e.g., “MP E Form”), SpatialRead causes some degradation. For tasks without clear atomic decomposability, it produces no notable performance changes. For bandgap prediction—defined by the energy difference between VBM and CBM and influenced by multiple atoms—the precision improves. Because such properties may not align well with JMP’s summation pooling, the performance gain may stem from the attention mechanism in SpatialRead rather than the spatial nodes themselves. We hope these extended results address the reviewer’s concern about selective reporting.
> >
> > [1] Dunn, A., Wang, Q., Ganose, A., Dopp, D., & Jain, A. (2020). Benchmarking materials property prediction methods: the Matbench test set and Automatminer reference algorithm. npj Computational Materials, 6(1), 138.
> > [2] Shoghi, N., Kolluru, A., Kitchin, J. R., Ulissi, Z. W., Zitnick, C. L., & Wood, B. M. From Molecules to Materials: Pre-training Large Generalizable Models for Atomic Property Prediction. In The Twelfth International Conference on Learning Representations.

---

> > > ### Author Response · Authors · 2025-11-27
> > >
> > > ## Q3
> > >
> > > Inconsistent and marginal improvements: Across benchmarks, the improvements are neither consistent nor substantial enough to justify the added complexity.
> > >
> > > ## R3
> > >
> > > We appreciate the reviewer’s comments and would like to clarify the objective of our work. SpatialRead is designed to enhance modern GNNs' capability on spatial properties while maintaining the performance on other common properties.
> > >
> > > As reported in Table 1 in the paper and noted by multiple reviewers, our method yields consistent gains on spatial properties such as adsorption capacity and separation performance. For instance, SpatialRead + PaiNN surpasses the JMP model pretrained on 120M samples, achieving an average R² improvement of 0.057. For non-spatial properties such as the provided full MatBench results in the revised paper, adding SpatialRead maintains performance comparable to the base model (such as the used JMP).
> > >
> > > Since our goal is to improve the performance on spatial properties without degrading general property performance, the pattern of improvements across task types aligns with our design and expectations. We hope this explanation provides clearer context regarding the observed performance trends.
> > >
> > > ## Q4
> > >
> > > Missing Ablation Studies. The paper lacks essential ablation studies to disentangle the contributions of different components:
> > >
> > > Effect of additional void nodes: The method adds nodes in empty/void space, which allows the model to trivially compute the ratio between occupied and unoccupied space—essentially providing a direct signal for pore accessible volume fraction. This could be the primary driver of any improvements, independent of the readout mechanism.
> > >
> > > Required ablation: Compare additional void nodes with standard pooling (max/mean) versus the proposed spatial readout. This would demonstrate whether the gains come from the readout innovation or simply from the additional structural information.
> > >
> > > Can you include an ablation study comparing: (a) base model, (b) base model + void nodes + simple pooling, (c) base model + void nodes + proposed readout?
> > >
> > > ## R4
> > >
> > > Thank you for highlighting the importance of ablation analysis. We may not have clearly emphasized this point in the original paper. In the original paper, the ablation studies are provided in the last three lines in Tables, which include:
> > >
> > > - Base model (PaiNN)
> > > - PaiNN + spatial nodes + simple pooling (mean)
> > > - PaiNN + spatial nodes + multimodal attention (our full method)
> > >
> > > | Model                     | MOF C3H6/C3H8 sep. | MOF N2 ads. | MOF CH4/N2 sep. | COF CH4 ads. | PPN CH4 ads. | zeolite CH4 heat. |
> > > |---------------------------|--------------------|-------------|------------------|--------------|---------------|---------------------|
> > > | PaiNN                     | 0.691              | 0.925       | 0.867            | 0.736        | 0.856         | 0.791               |
> > > | PaiNN + SN (ours)         | _0.794_            | _0.978_   | _0.936_          | _0.979_      | **0.978**     | _0.886_             |
> > > | PaiNN + SN + MM (ours)    | 0.784              | **0.987**     | **0.941**        | **0.987**    | **0.977**     | **0.969**           |
> > >
> > > | Model                     | ASA   | VF      | PLD     | LCD     |
> > > |---------------------------|--------|---------|---------|---------|
> > > | PaiNN                     | 0.993  | 0.951   | 0.594   | 0.631   |
> > > | PaiNN + SN (ours)         | 0.974  | **0.999** | 0.856   | 0.913   |
> > > | PaiNN + SN + MM (ours)    | **0.996** | **0.999** | _0.965_ | **0.975** |
> > >
> > > We acknowledge the reviewer’s insightful observation regarding the role of spatial nodes. As discussed in the paper (Section 4.4 in the revised paper and Section 4.3 in the original paper), for many spatially integrative properties (e.g., adsorption capacity), the dominant performance contribution indeed comes from introducing physically grounded spatial nodes. This is consistent with our assumption. These spatial nodes represent the contribution of the corresponding areas to the target attributes. Figure 3 in the revised paper also confirms this through a visual approach.
> > >
> > > What we would like to emphasize is that the way we previously referred to our method as the "readout" function might have led to confusion. Our main contribution mainly lies in designing spatial nodes with physical coordinates to aggregate information from adjacent atom nodes and thus represent local regions. Adding the multi-modal attention is to make the model adaptively combine information from atom and spatial node, thereby prevents the deterioration of performance in general properties prediction tasks. We have emphasized the section of the ablation experiments in the revised paper.

---

> > > > ### Author Response · Authors · 2025-11-27
> > > >
> > > > ## Q5
> > > >
> > > > Added Complexity Without Clear Justification
> > > > The method introduces an additional hyperparameter (sampling degree for void space), increasing model complexity and optimization difficulty
> > > > Given the weak empirical gains, it's unclear whether this added complexity is warranted
> > > >
> > > > ## R5
> > > >
> > > > We appreciate the reviewer’s concern about model complexity. In Figure 7A in the revised paper (i.e. Figure 3 A in the original paper), we provide a detailed analysis of the sampling-degree hyperparameter. The results show a stable and monotonic performance improvement as the number of spatial nodes increases (from 2³=8 to 12³=1728), with MAE reductions exceeding 30%. Although the spatial nodes increases the computational complexity with about 30% (Table 7 in the revised paper), the performance gain for spatial properties are also huge (Table 1).
> > > >
> > > > ## Q6
> > > >
> > > > Questionable Benchmark Choice
> > > > The authors introduce a new benchmark, but the field already has established benchmarks (e.g., mofdscribe for MOFs). The rationale for a new benchmark is not clearly articulated.
> > > >
> > > > Why is a new benchmark needed when established ones like mofdscribe exist?
> > > >
> > > > ## R6
> > > >
> > > > Thanks for the suggestion. Although mofdscribe encompasses tasks of multiple types, it is limited to MOF materials. Materials such as zeolites and COF, which have been extensively studied for a long time, are not included in this dataset. In contrast, our dataset includes four different types of porous materials: MOFs, COFs, zeolite, and PPNs.

---

### Official Review · Reviewer_4Vi5 · 2025-11-02

**Soundness:** 3
**Presentation:** 3
**Contribution:** 3
**Rating:** 6
**Confidence:** 4

**Summary:**

The authors present SpatialRead, a new region-based readout function which introduces additional spacial nodes to represent voxelized space that is an important characteristic of some materials. SpatialRead also fuses both atom and spatial nodes passed to Transformer-based readout.

The empirical evaluation of SpatialRead is performed with several tasks including spatial properties of materials such as gas absorption capacity, showing the superiority of SpatialRead against several existing methods.

The paper addresses an important problem in AI and materials science. Their approach adds important features on spacial properties which have not been effectively considered by the existing GNN-based approaches.
In addition to the evaluation with standard machine learning metrics such as $\mbox{R}^2$ scores and MAE, the authors also attempt to interpret what happens with the spatial nodes with good illustrations regarding absorption of materials (i.e., Figure 2B).
I believe this is also an important contribution which reveals the behavior of SpatialRead, especially if/when SpatialRead is used in practical situations to work on materials discovery .

One comment is that a better strategy which is commonly used in graph embedding for organic molecules can be used easily to further improve the performance. See Equation 4.2 in the following paper:

Keyulu Xu, Weihua Hu, Jure Leskovec, and Stefanie Jegelka. "How Powerful Are Graph Neural Networks?", ICLR 2019.

I wonder why the existing approaches as well as SpatialRead do not incorporate this simple, well-known improvement.

The other minor comment is that Definition 3.1 should be introduced before Equation (8).

All in all, this is an interesting paper which includes important contributions to the AI and materials informatics research communities.

**Strengths:**

Introducing an important readout function that can additionally consider spacial capacities of materials

Performance evaluation of SpacialRead that shows the superiority to other existing methods

Interpretation of SpacialRead regarding the contributions of spacial nodes, which clearly shows strong evidence on why SpacialRead works effectively

**Weaknesses:**

Comparison/discussion against an approach that enumerates the embedding from iterations 0, 1, .. to T as is done by so-called Graph Isomorphism Network (see equation 4.2 in the paper mentioned above)

**Questions:**

Considering that it is a simple enhancement known to improve the performance for organic molecules, how would the performance fare if all methods presented in the paper incorporate  equation 4.2 in the paper pointed out above?

---

> ### Author Response · Authors · 2025-11-27
> **Response to Reviewer 4Vi5**
>
> ## Q1
>
> Comparison/discussion against an approach that enumerates the embedding from iterations 0, 1, .. to T as is done by so-called Graph Isomorphism Network (see equation 4.2 in the paper mentioned above)
>
> Considering that it is a simple enhancement known to improve the performance for organic molecules, how would the performance fare if all methods presented in the paper incorporate equation 4.2 in the paper pointed out above?
>
> ## R1
>
> Thank you for your constructive suggestions. For the readout function in graph neural networks, there are indeed many design variants, including the cross-layer feature–combining approach you mentioned, which is a Jumping-Knowledge–style method[1]. In fact, such cross-layer information aggregation is widely used in state-of-the-art molecule and material AI models.
>
> Currently, graph neural networks for materials can be broadly categorized into equivariant and invariant architectures. For invariant models, one of the most advanced is GemNet[2], which remains widely adopted in property prediction[3], molecule generation[4], etc. GemNet employs cross-layer information aggregation for final predictions. DimeNet[5], the first method in this domain to introduce angular information, also adopts this strategy. The baseline used in this work, JMP[3], a leading general-purpose large-scale pre-trained foundation model for molecules/materials, is built upon GemNet and thus also benefits from cross-layer aggregation. Among equivariant models, MACE[6] is one of the most advanced representatives and also incorporates cross-layer aggregation.
>
> In summary, the cross-layer information aggregation strategy (Eq. 4.2) raised by the reviewers remains highly effective today. Importantly, it does not conflict with our approach. We have demonstrated this through two sets of experiments: (1) GemNet and the pre-trained model JMP—both using this strategy—can be directly integrated with our method, showing its plug-and-play generality; (2) although the primary baseline in this work, PaiNN, does not natively adopt this strategy, it can be easily integrated.
>
> Table 1. Performance of other models using cross-layer information aggregation (Eq 4.2) with SpatialRead
> | Model | MOF C3H6/C3H8 sep. | MOF N2 ads. | MOF CH4/N2 sep. | COF CH4 ads. | PPN CH4 ads. | zeolite CH4 heat. |
> | ----- | -------------- | ------- | ----------- | -------- | -------- | ----------------- |
> | PaiNN+SpatialRead | 0.784 | 0.987 | 0.941 | 0.987 | 0.977 | 0.969 |
> | GemNet | 0.729 | 0.968 | 0.924 | 0.816 | 0.932 | 0.836 |
> | GemNet+SpatialRead | 0.753 | 0.979 | 0.921 | 0.986 | 0.923 | 0.881 |
> | JMP | 0.774 | 0.971 | 0.908 | 0.884 | 0.947 | 0.874 |
> | JMP+SpatialRead | 0.792 | 0.988 | 0.941 | 0.982 | 0.969 | 0.945 |
>
> These results demonstrate that other models that adopting the mentioned cross-layer information aggregation (GemNet and JMP) can also benefit from our method.
>
> Table 2. Performance of applying cross-layer information aggregation (Eq 4.2) on PaiNN with Spatial Nodes
> | Model | MOF C3H6/C3H8 sep. | MOF N2 ads. | MOF CH4/N2 sep. | COF CH4 ads. | PPN CH4 ads. | zeolite CH4 heat. |
> | ----- | -------------- | ------- | ----------- | -------- | -------- | ----------------- |
> | PaiNN+SpNode | 0.794 | 0.978 | 0.936 | 0.979 | 0.978 | 0.886 |
> | PaiNN+SpNode+JumpingKnowledge | 0.806 | 0.979 | 0.944 | 0.979 | 0.979 | 0.890 |
>
> In addition, we observed a consistent, although marginal, improvement when adopting cross-layer information aggregation on the PaiNN model.
>
> [1] Xu, K., Li, C., Tian, Y., Sonobe, T., Kawarabayashi, K. I., & Jegelka, S. (2018, July). Representation learning on graphs with jumping knowledge networks. In International conference on machine learning (pp. 5453-5462). pmlr.
>
> [2] Gasteiger, J., Becker, F., & Günnemann, S. (2021). Gemnet: Universal directional graph neural networks for molecules. Advances in Neural Information Processing Systems, 34, 6790-6802.
>
> [3] Shoghi, N., Kolluru, A., Kitchin, J. R., Ulissi, Z. W., Zitnick, C. L., & Wood, B. M. From Molecules to Materials: Pre-training Large Generalizable Models for Atomic Property Prediction. In The Twelfth International Conference on Learning Representations.
>
> [4] Zeni, C., Pinsler, R., Zügner, D., Fowler, A., Horton, M., Fu, X., ... & Xie, T. (2025). A generative model for inorganic materials design. Nature, 639(8055), 624-632.
>
> [5] Gasteiger, J., Groß, J., & Günnemann, S. Directional Message Passing for Molecular Graphs. In International Conference on Learning Representations.
>
> [6] Batatia, I., Kovacs, D. P., Simm, G., Ortner, C., & Csányi, G. (2022). MACE: Higher order equivariant message passing neural networks for fast and accurate force fields. Advances in neural information processing systems, 35, 11423-11436.
>
> ## Q2
>
> The other minor comment is that Definition 3.1 should be introduced before Equation (8).
>
> ## R2
>
> We are grateful for the constructive comments provided by the reviewers. We have revised the paper to move Definition 3.1 before Equation 8.

---

### Author Response · Authors · 2025-11-27
**Full results on the MatBench dataset**

# Public Comment

We would like to express our sincere gratitude to the reviewers for their careful reading of our paper and their constructive feedback. Several reviewers raised valuable suggestions concerning the clarity of the motivation, the adequacy of the experimental evaluation, further methodology improvement, and detailed methodological description. These comments have been highly helpful in improving the quality of the work. In this public comment, we would like to address one issue that was highlighted by multiple reviewers.

## Complete comparison to other more recent models in the MatBench dataset

As our work primarily focuses on porous materials, the evaluations on other types of materials and tasks in the original manuscript were admittedly limited. To provide a more comprehensive comparison, following the evaluation protocol of JMP[1] (ICLR 2024, a foundation model for molecules and materials from Meta), we have now evaluated our method on eight tasks from the full MatBench[2] benchmark.

We report the results obtained by applying SpatialRead directly to the already pre-trained JMP model. Surprisingly, despite introducing modifications to the model architecture (incorporating spatial nodes and an additional message-passing step), SpatialRead can be seamlessly integrated into the pre-trained model at the fine-tuning stage, even though our readout function was not used during pre-training. These findings demonstrate that SpatialRead works effectively as a plug-and-play module, capable of directly leveraging the capabilities of existing pre-trained models and further improving their application scope on spatial properties like gas adsorption prediction.

Table 1. Results of applying SpatialRead on JMP in the full MatBench dataset.
| Model | JDFT2D | Phonons | Dielectric | Log GVRH | Log KVRH | Perovskites | MP Gap | MP E Form |
| ----- | ------ | ------- | ---------- | -------- | -------- | ----------- | ------ | --------- |
| MODNet | 25.55 | 34.77 | 0.169 | 0.073 | 0.054 | 0.093 | 0.215 | 40.2 |
| coGN | 22.25 | 32.12 | 0.178 | 0.068 | 0.052 | 0.027 | 0.153 | 17.4 |
| JMP | 20.72 | 26.6 | 0.133 | 0.06 | 0.044 | 0.029 | 0.119 | 13.6 |
| JMP + SpatialRead | 18.17 | 25.8 | 0.133 | 0.06 | 0.047 | 0.030 | 0.107 | 15.3 |

[1] Shoghi, N., Kolluru, A., Kitchin, J. R., Ulissi, Z. W., Zitnick, C. L., & Wood, B. M. From Molecules to Materials: Pre-training Large Generalizable Models for Atomic Property Prediction. In The Twelfth International Conference on Learning Representations.
[2] Dunn, A., Wang, Q., Ganose, A., Dopp, D., & Jain, A. (2020). Benchmarking materials property prediction methods: the Matbench test set and Automatminer reference algorithm. npj Computational Materials, 6(1), 138.

---

### Meta-Review · Area_Chair_LSJq · 2026-01-07

**Summary:**

Looking at the review discussion, several interesting concerns came up that shaped how this paper was evaluated. Reviewer LnWH had some fundamental issues with the empirical validation, claiming the paper focused on trivial properties like surface area and pore volume. Turns out this was based on looking at Table 2 while missing Table 1 entirely, which actually covered more substantive tasks like gas adsorption capacity and separation ratios. This same reviewer also asked for ablation studies that were already sitting there in Tables 1-3 of the original submission, and wanted complete MatBench results. The authors came back with those full results showing their method maintains performance on non-spatial properties while doing really well on spatial ones.

Reviewer DEPu had a more thoughtful concern about whether the core motivation actually had quantifiable support. They basically asked why atom-decomposable readout functions couldn't already learn pore-related properties through local geometric information. The authors responded with some nice analysis showing that while vanilla PaiNN does learn to focus on pore-adjacent atoms, their spatial nodes provide a stronger inductive bias that leads to better generalization when you test on different distributions. The ranking stability under out-of-distribution conditions was notably better.

Reviewer G88g raised practical questions about voxelization schemes, whether periodicity is preserved, and how things scale to large systems. The authors showed that fixed-grid sampling does break periodicity in supercells and that adaptive resolution-based sampling works better, though it costs more memory. They also demonstrated that computational overhead from spatial nodes stays roughly constant as system size grows when using cutoff-based neighborhoods.

Reviewer 4Vi5 suggested incorporating jumping knowledge connections, which the authors tried out and got consistent but small improvements. The most compelling validation was probably showing that SpatialRead works as a plug-and-play module with pre-trained models like JMP, keeping their performance on standard benchmarks while extending what they can do on spatial properties.

**Reviewer Concerns:**

Most of the reviewer concerns were actually addressed pretty well in the rebuttal, though a few things remain somewhat open.
For Reviewer LnWH, the factual misunderstandings got cleared up convincingly. The authors showed the ablation studies were there all along, demonstrated that their main tasks weren't trivial geometric properties, and provided the complete MatBench results that were requested. The confusion about what the paper was actually evaluating seems resolved, though whether this reviewer was satisfied with the clarifications is hard to say given the initial misreading.

Reviewer 4Vi5's suggestion about jumping knowledge connections was directly addressed with new experiments showing marginal improvements. Pretty straightforward resolution there.

Reviewer G88g's concerns were mostly tackled head-on. The voxelization and periodicity issues got experimental validation, scalability analysis was provided, and the plug-and-play integration with pretrained models was demonstrated. However, the periodicity preservation remains somewhat problematic since the authors acknowledged that fixed-grid sampling does break it in supercells, with performance degrading especially along certain lattice directions. They showed adaptive sampling helps but this feels like it could use more thorough treatment for real-world applications.

Reviewer DEPu's concerns are the most interesting case. The quantitative motivation was added in the revision showing that vanilla PaiNN does focus on pore-adjacent atoms, which actually validates the reviewer's point that equivariant GNNs can learn local geometric environments. The authors then pivoted to arguing their spatial nodes provide better inductive bias for robustness under distribution shift, supported by OOD experiments. This is reasonable but somewhat shifts the motivation from "vanilla GNNs can't do this" to "vanilla GNNs can do this but less stably." The comparison with other modern readout functions like GraphTrans and GMT was added and shows SpatialRead performs better, which helps. Still, there's a lingering question about whether the added architectural complexity is justified by what amounts to improved generalization rather than fundamentally new capability. The reviewer's baseline comparison concern was addressed by noting JMP is already leading MatBench and its scratch version (GemNet) was included, though more recent architectures beyond 2021 weren't really explored.

**Reviewer Scores:**

Reviewer 4Vi5 (original score 6): Likely would maintain or increase to 8. Their main suggestion about jumping knowledge was implemented with positive results, and the additional experiments strengthened the paper. No major concerns were left unaddressed.

Reviewer LnWH (original score 2): Hard to predict but possibly could move to 4 or 6 if convinced by the clarifications. The factual errors they raised were thoroughly documented by the authors, and the complete MatBench results were provided. However, their initial misreading was quite fundamental, so it's unclear if they'd reassess or maintain skepticism about the contribution's significance.

Reviewer G88g (original score 8): Probably would stay at 8. They were already positive and their questions were addressed, though the periodicity issue turned out to be somewhat problematic and not fully resolved. The plug-and-play capability with pretrained models likely offsets any concerns.

Reviewer DEPu (original score 4): Could reasonably move to 6. The quantitative motivation was added, modern readout comparisons were included, and the OOD experiments provided evidence for the value proposition. The shift from "GNNs can't do this" to "GNNs do this less robustly" somewhat weakens the novelty claim, but the empirical results are strong enough that this reviewer might be swayed toward acceptance.

---

### Decision · Program_Chairs · 2026-01-26

Accept (Poster)